# A systematic pipeline for classifying bacterial operons reveals the evolutionary landscape of biofilm machineries

**Cedoljub Bundalovic-Torma**[1,2¤a], **Gregory B. Whitfield**[1,2¤b], **Lindsey S. Marmont**[1,2¤c], **P. Lynne Howell**[1,2], **John Parkinson**[1,2,3]*

**1** Program in Molecular Medicine, The Hospital for Sick Children, Toronto, Ontario, Canada, **2** Department of Biochemistry, University of Toronto, Toronto, Ontario, Canada, **3** Department of Molecular Genetics, University of Toronto, Toronto, Ontario, Canada

¤a Current address: Department of Cell & Systems Biology, University of Toronto, Toronto, Ontario, Canada
¤b Current address: Département de Microbiologie, Infectiologie et Immunologie, Université de Montréal, Montréal, QC, Canada
¤c Current address: Department of Microbiology, Harvard Medical School, Boston, Massachusetts, United States of America
* john.parkinson@utoronto.ca

**Data Availability Statement:** All relevant data are within the manuscript and its Supporting Information files.

## Abstract

In bacteria functionally related genes comprising metabolic pathways and protein complexes are frequently encoded in operons and are widely conserved across phylogenetically diverse species. The evolution of these operon-encoded processes is affected by diverse mechanisms such as gene duplication, loss, rearrangement, and horizontal transfer. These mechanisms can result in functional diversification, increasing the potential evolution of novel biological pathways, and enabling pre-existing pathways to adapt to the requirements of particular environments. Despite the fundamental importance that these mechanisms play in bacterial environmental adaptation, a systematic approach for studying the evolution of operon organization is lacking. Herein, we present a novel method to study the evolution of operons based on phylogenetic clustering of operon-encoded protein families and genomic-proximity network visualizations of operon architectures. We applied this approach to study the evolution of the synthase dependent exopolysaccharide (EPS) biosynthetic systems: cellulose, acetylated cellulose, poly-β-1,6-N-acetyl-D-glucosamine (PNAG), Pel, and alginate. These polymers have important roles in biofilm formation, antibiotic tolerance, and as virulence factors in opportunistic pathogens. Our approach revealed the complex evolutionary landscape of EPS machineries, and enabled operons to be classified into evolutionarily distinct lineages. Cellulose operons show phyla-specific operon lineages resulting from gene loss, rearrangement, and the acquisition of accessory loci, and the occurrence of whole-operon duplications arising through horizonal gene transfer. Our evolution-based classification also distinguishes between PNAG production from Gram-negative and Gram-positive bacteria on the basis of structural and functional evolution of the acetylation modification domains shared by PgaB and IcaB loci, respectively. We also predict several *pel*-like operon lineages in Gram-positive bacteria and demonstrate in our companion paper

**Funding:** JP and CB-T were supported by grants from the Natural Sciences and Engineering Research Council (RGPIN-2014-06664 & RGPIN-2019-06852; http://www.nserc-crsng.gc.ca/) and the National Institutes of Health (NSERC; R21AI126466; https://www.nih.gov/). This work was also supported in part by grants from the Canadian Institutes of Health Research (CIHR; http://www.cihr-irsc.gc.ca/) (MOP 43998 and FDN154327 to PLH). PLH is a recipient of a Canada Research Chair (http://www.chairs-chaires.gc.ca/home-accueil-eng.aspx). GBW and LSM have been supported by graduate scholarships from NSERC. GBW has been supported by a graduate scholarship from Cystic Fibrosis Canada (https://www.cysticfibrosis.ca/). LSM has been supported by graduate scholarships from the Ontario Graduate Scholarship Program (https://osap.gov.on.ca/OSAPPortal/en/A-ZListofAid/PRDR019245.html), and The Hospital for Sick Children Foundation Student Scholarship Program (http://www.sickkids.ca/Research/StudentandFellowResources/RTC/Training-Programs/restracomp/index.html). Computing resources were provided by the SciNet HPC Consortium; SciNet is funded by: the Canada Foundation for Innovation (https://www.innovation.ca/) under the auspices of Compute Canada (https://www.computecanada.ca/); Ontario Research Fund–Research Excellence (https://www.ontario.ca/page/ontario-research-fund) and the University of Toronto (https://www.utoronto.ca/). The funders had no role in study design, data collection and analysis, decision to publish, or preparation of the manuscript.

**Competing interests:** The authors have declared that no competing interests exist.

(Whitfield *et al* PLoS Pathogens, in press) that *Bacillus cereus* produces a Pel-dependent biofilm that is regulated by cyclic-3',5'-dimeric guanosine monophosphate (c-di-GMP).

## Author summary

In bacterial genomes, biological processes are frequently encoded by neighbouring co-transcribed genes, termed operons. In addition, operon-associated genes often belong to distinct evolutionary families with diverse biological functions. Studying the evolution of bacterial operons provides valuable insight into understanding the biological significance of genes involved in environmental adaptation. To date, no systematic approach has been devised to examine both the complex evolutionary relationships of operon encoded genes and the evolution of operon organization as a whole. To address this challenge, we developed an integrative method to study operon evolution by combining phylogenetic tree based clustering and genomic-context networks. We applied this method to perform the first systematic survey of all known synthase-dependent exopolysaccharide biosynthetic machineries, demonstrating the generalizability of our approach for operons of diverse size, protein family composition, and species distribution. Our method identified distinct biofilm operon clades across phylogenetically diverse bacteria, that result from gene rearrangement, duplication, loss, fusion, and horizontal gene transfer. We found different evolutionary trajectories for Gram-negative and Gram-positive PNAG biofilm production machineries, and in a companion paper (Whitfield *et al* PLoS Pathogens, in press) present experimental validation that the Pel polysaccharide is produced by a Gram-positive bacterium.

## Introduction

The generation of novel genomes through next generation sequencing is creating a wealth of opportunities for understanding the evolution of biological systems. A key challenge is the development of robust and systematic approaches that allow genes to be classified into functional categories and which are also capable of inferring evolutionary relationships. In bacterial genomes, functionally-related genes corresponding to metabolic pathways or protein complexes are often encoded by neighbouring co-regulated and co-transcribed loci, termed an operon. Computational prediction of operons based on the conservation of short inter-genetic distances found between homologous genes across phylogenetically diverse bacteria has been frequently used to predict the biological roles of neighbouring uncharacterized genes [1–6]. Such approaches are valuable for computational inference of gene function in experimentally uncharacterized organisms and facilitate comparative genomics of adaptive traits across phylogenetically diverse bacteria.

Analyzing patterns of sequence divergence within each gene yields insights into species-specific functionalities. However, genes in an operon do not function in isolation but typically form parts of higher-order, biological modules (*e.g.* protein complexes or metabolic pathways). Consequently, analysing evolutionary events in an operonic context provides additional opportunities to better infer functional relationships. For example, while sequence divergence has the potential to impact the function of a single gene, evolutionary events that alter operon structure (e.g. rearrangements, duplications, gains and losses) have the potential to dramatically alter the overall function of the operon [7,8].

Due to the lack of a systematic framework, very few studies have attempted to examine the influence of evolutionary events on operon structure [9,10]. Phylogenetic-tree based classification of 197 ATP binding motif sequences associated with operon-encoded bacterial ATP-binding cassette (ABC) transporters was successful in resolving two evolutionarily distinct transporter clades associated with import and export functions [11]. Gene duplications have been shown to play an important role in driving protein superfamily expansion and are positively correlated with bacterial genome size [12]. Duplications have been found to be associated with biological processes associated with environmental adaptation of species clusters [13], such as outer membrane polysaccharide export proteins involved in capsule biosynthesis [14,15], and amplification of beta-lactamase enzymes associated with increased antibiotic resistance [16]. The study of co-localized "gene blocks" across bacteria has also shown that gene duplication, loss, and rearrangement play important roles in shaping the large-scale organization of bacterial genomes [10]. Key to these analyses is the use of a rigorous and systematic approach for assigning genes into evolutionarily related 'families' that are likely to share similar functions. However, the inference of biological function based on sequence similarities of genes or proteins is often complicated by functional divergence arising through recent gene duplication events. A variety of metrics have been employed for determining the relatedness of genes and their protein products from which groups (i.e. clustering) can be defined. These metrics include: evolutionary distances derived through the construction of phylogenetic trees [17–19]; global protein sequence similarities [20–22]; and shared sequence features such as conserved amino acids at specific sites or shared amino acid subsequences, which define motifs or structural domains [23,24]. The aim of these approaches is to automatically resolve large protein families comprising potentially thousands of genes into a smaller number of clusters defining evolutionarily related subfamilies with similar biological roles.

An additional challenge faced by clustering methodologies is defining which set of clusters result in an optimal partitioning of the underlying data. To help guide such partitioning, a variety of cluster validation approaches have been devised. These are broadly divided into two categories: external-validation and internal-validation, based on whether previous information is available for the data being clustered [25]. For example, methods developed for classifying orthologous genes (*i.e.* those related by common ancestry) and paralogous genes (*i.e.* those emerging from duplication after a speciation event) rely on internal-validation tests. In one such approach, internal branch lengths between one-to-many homologous gene relationships are compared between two species [26]. In an alternative approach, clustering is employed to define "triangles" of proteins with significant sequence similarity occurring between three distinct species [20,27]. However, to distinguish the finer-scale evolutionary relationships occurring within an orthologous group or gene family, phylogenetic tree construction is required. Such methods have typically focused on well-characterized biological systems, *e.g.* homologs of the bacterial flagellar and type III secretion system subunits [28] and diverse systems associated with the type IV filament superfamily [29], which have utilized an external-validation approach for defining functionally distinct phylogenetic clades.

Here we build on these methods and present a framework for the systematic classification and analysis of diverse gene families in the context of operons. Focusing on synthase-dependent exopolysaccharide (EPS) biosynthetic machineries, we use our framework to explore how gene divergence in combination with duplication, loss, and rearrangement events have shaped the evolution of EPS operons, and may have influenced the biofilm producing capabilities of evolutionarily diverse bacteria.

EPS are an important component of bacterial biofilms that not only ensure survival in response to limited nutrient availability, but are also involved in antibiotic tolerance, immune evasion and serve as virulence factors in many clinically relevant pathogens [30–32]. Distinct

mechanisms have been identified in the production of bacterial EPS, including the well-characterized Wzx/Wzy and ABC transporter-dependent pathways [33], and synthase-dependent systems [34].

Typically, Gram-negative synthase-dependent EPS systems are organized as discrete operons comprised of genes encoding: 1) an inner membrane associated polysaccharide synthase; 2) a regulatory domain or co-polymerase subunit responsible for binding the intracellular signaling molecule cyclic-3',5'-dimeric guanosine monophosphate (c-di-GMP); 3) periplasmic polysaccharide modification enzymes; and 4) a periplasmic tetratricopeptide repeat (TPR) domain coupled with an outer membrane pore [34]. This operonic organization allows bacteria to acquire complete EPS functionality through discrete lateral gene transfer events and may act as a key driver in niche adaptation [35]. To date five synthase-dependent EPS have been identified: cellulose, acetylated cellulose, poly-β-1,6-N-acetyl-D-glucosamine (PNAG), alginate and the Pel polysaccharide. While much interest has focused on the molecular basis of biofilm formation, these systems have been characterized for only a relatively limited set of bacterial species. Consequently, little is known concerning how these systems have evolved. Of interest is how mechanisms such as gene divergence, duplication, acquisition, loss, and rearrangement of EPS operons have contributed to bacterial adaptation to diverse environments, and from a human health perspective, contributed to a pathogen's ability to infect and cause disease. While a previous survey of cellulose EPS machineries has been reported [36], a comprehensive systematic analysis of all EPS machineries is lacking.

In this study, we describe a phylogenetic tree-based clustering method for defining protein sequence subfamilies and apply it to study the evolutionary relationships of operons. This method was employed for the systematic classification of EPS operons predicted from a survey of over a thousand bacterial genomes. Applying a graphical visualization approach, we demonstrate that phylogenetic clustering enables the resolution of discrete EPS operon clades which differ in their organization from experimentally characterized operons, providing valuable insights toward further understanding the roles of gene duplication, rearrangement, and loss/absence in the evolution of biofilm production between phylogenetically diverse species from distinct environmental niches. For example, we demonstrate the biological implications of operon evolution that has been shaped by horizontal gene transfer (HGT) and subsequent divergence, for two cellulose operon clades among Proteobacteria which correspond to the production of cellulose polymers with different structural organizations. Furthermore, we note that most of our operon predictions are novel and demonstrate the value of applying computational predictions to guide the discovery of EPS production in previously uncharacterized species. We highlight an example for Pel production, which was initially identified and characterized in *Pseudomonas aeruginosa* [37] and other Gram-negative bacteria. Our approach identified several *pel*-like operons in some *Bacillus* spp. and other Gram-positive bacteria, which appeared to be regulated by the intracellular signaling molecule c-di-GMP. In our companion paper (Whitfield *et al* PLoS Pathogens, in press) we experimentally validate these findings by demonstrating the production of Pel by the Gram-positive *Bacillus cereus* ATCC 10987 and its regulation by c-di-GMP.

## Results

### A systematic survey of bacterial EPS operons reveals EPS systems across bacteria of diverse lifestyles and environmental niches

A schematic overview of the pipeline we used for classifying bacterial operons is provided in **Fig 1A**. The process begins with a systematic survey of all five previously characterized synthase-dependent EPS systems (cellulose, acetylated cellulose, PNAG, Pel, and alginate) (**S1**

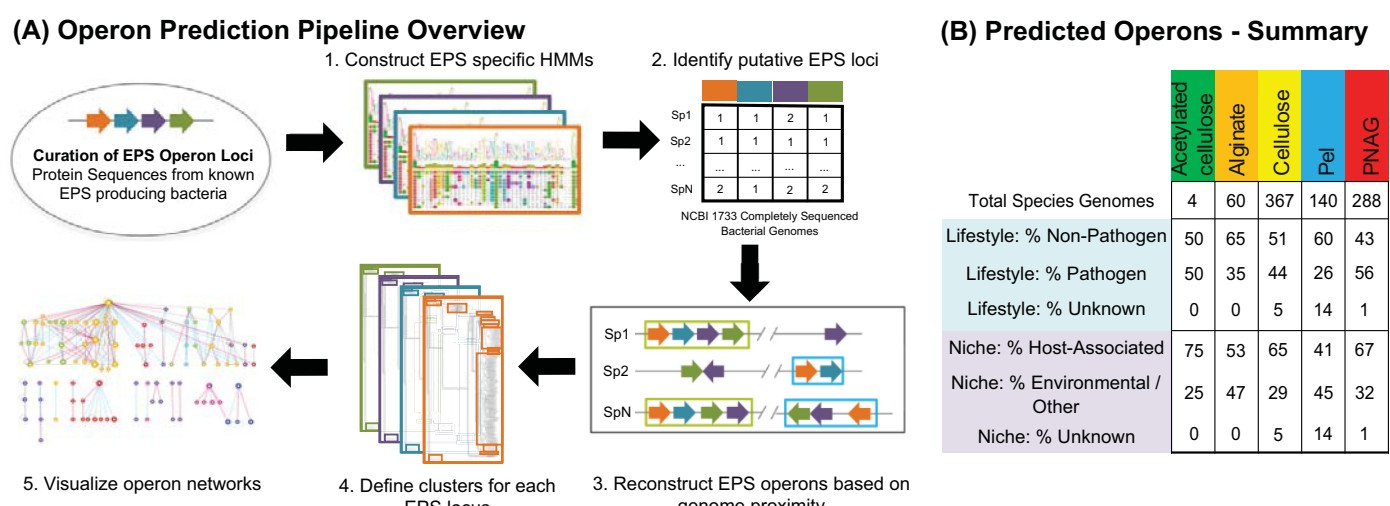

**Fig 1. Summary of predicted bacterial EPS operons.** (A) Overview of the main steps of the EPS operon prediction pipeline: 1) *De novo* construction of reference operon locus HMM models; 2) Sequence homology searches against 1733 fully sequenced bacterial genomes retrieved from NCBI; 3) Prediction of putative EPS operons through genomic-proximity reconstruction of significant locus hits; 4) Definition of clusters defining evolutionary relationships for each EPS locus; 5) Integration of operon and cluster predictions to visualize operon networks. (B) Number of predicted EPS operons and percentages of species summarized by bacterial lifestyle (pathogen, non-pathogen, unknown) and corresponding niche (host-associated, environmental/other, unknown). (C) Percentage of evolutionary events associated with EPS operons: Locus loss (core EPS operon loci not detected by HMM searches); Locus rearrangement (EPS operons featuring locus orderings that differ from the canonical operon for that type–S1 Table); Locus duplication (defined by two loci possessing a significant match to the same EPS HMM within the same operon); Operon duplication, defined as a genome encoding two copies of the same type of EPS system, separated by greater than 10 kb; Locus fusion, loci possessing significant matches to multiple EPS HMMs. (D) A cladogram based on 16S rRNA sequences illustrating the phylogenetic distribution of predicted EPS operons. Branches are coloured according to taxonomic class and piecharts represent the proportion of EPS operon types identified for each clade. Leaf labels represent the major taxonomic families found in each clade, along with the number of genomes represented. 16S rRNA sequences were retrieved from full genome sequences using Barrnap (https://github.com/tseemann/barrnap), clustered using CD-HIT [94] at 95% sequence identity and aligned using MUSCLE [92] for phylogenetic tree construction with Fasttree [103] and subsequent visualization with iTOL [104]. (E) An upset plot (generated using the R (https://www.r-project.org) package UpSetR [105]) representing the number of different EPS operon combinations identified across bacterial genomes.

& S2 Tables) through an iterative hidden Markov-model (HMM) -based search strategy and subsequent genomic-proximity based reconstruction of 1733 complete reference and representative bacterial genomes (downloaded April 20, 2015—see Methods). We identified 407 cellulose, 321 PNAG, 146 Pel, 64 alginate, and 4 acetylated cellulose EPS "operons" defined as comprising at least: 1) a polysaccharide synthase subunit; and 2) one additional locus involved in EPS modification or transport as defined previously [30] (S3 Table). These could be allocated to 367, 288, 140, 60 and 4 different bacterial species, respectively (Fig 1). We note that for all previously characterized EPS producing species included in this set of fully sequenced genomes, we successfully detected an operon corresponding to the type of EPS produced (S4 Table). Furthermore, for experimentally characterized species lacking a fully sequenced genome, we also identified several closely related strains possessing an EPS operon identical to the characterized strain, providing a valuable resource for further experimental validation. PNAG was significantly enriched in pathogen genomes (161/288–56%; Chi-squared test p-value = 2.05e-09). Conversely, Pel (84/140–60%; Chi-squared test p-value ~ 1.83e-4), alginate (39/60–65%; Chi-squared test p-value ~ 3.5e-2) and cellulose (187/367–51%; Chi-squared test p-value = 2.05e-14) were significantly enriched in non-pathogen genomes (Fig 1C and S1 Fig). Interestingly, both cellulose and PNAG operons were significantly associated with genomes with host-associated lifestyles (Chi-squared p-values ~ 1.05e-9 and ~1.48e-12, respectively). From a phylogenetic perspective, each EPS system was well represented by Proteobacteria, with cellulose, PNAG and Pel additionally featuring operons from Bacilli and Clostridia, which to our knowledge have not been previously reported (Fig 1D). Further, we note that Pel operons exhibited the greatest diversity of bacterial families (Shannon index of bacterial families– 2.74) with representation in Thermotogae, Actinobacteria and Rubrobacteria, among others. While most genomes contain only a single synthase-dependent EPS system, we observed many instances of co-occurrence (Fig 1E), with cellulose and PNAG systems being the most common combination (83 genomes), followed by alginate and Pel (20 genomes). Notably, all species possessing three systems were *Pseudomonas* spp., e.g.: *Pseudomonas protegens* strains Pf-5 and CHA0 (alginate, Pel and PNAG); *Pseudomonas fluorescens* SBW25 and *Pseudomonas* sp. TKP (acetylated cellulose, alginate, and PNAG).

## Evolution of EPS operons is driven by gene duplication, loss and rearrangements

The processes underlying EPS operon evolution across diverse bacterial phyla are poorly understood. We examined how operon organization is influenced by the following evolutionary events that are likely to affect EPS production capabilities among bacteria: 1) single locus or whole operon duplications, which could lead to dosage effects and alter the level of EPS

modification or export; 2) locus losses, that may indicate a reduction or loss in EPS production or modification, or may suggest supplementation of the lost function with a novel gene; 3) operon rearrangements which may affect the regulation of EPS production through the order of expression of individual EPS system components; and, 4) gene-fusions, resulting in enhanced co-expression of interacting subunits.

For each set of predicted EPS operons, the resulting number of operon evolutionary events was defined relative to the locus composition and order of reference Gram-negative experimentally characterized operons [30,38–41]. In contrast to previous studies of operon evolution [28,29], we use the term evolutionary events to refer to key changes that define distinct organizations between evolutionarily distinct operon clades and not divergence of operons from an ancestral state. With the exception of acetylated cellulose, locus losses were found to be the most frequent event (~46% of predicted operons lacked one or more reference loci), and occurred with the greatest frequency for Pel which exhibited an average loss of 2.6 loci lost per operon (**S5 Table**). Among all EPS systems the majority of locus losses were associated with the outer membrane pore encoding loci (441 / 993–44% of all locus loss events identified) among Gram-positive species (**S5 Table**), consistent with the lack of an outer membrane bilayer in Gram-positive cell envelope architectures. Operon rearrangements were the next most frequent evolutionary events (~ 39%), largely associated with cellulose operons [36] (327 / 407–80%). Focusing on duplication events, within-operon loci duplications tended to be more common than whole-operon duplications (2 or more core EPS loci identified $> = 1$ kb apart), with the exception of cellulose operons (29 whole operon duplications compared to 24 loci duplications). All duplicated operons were found to be separated by at least 10 kb, suggesting they may have been acquired through HGT rather than tandem duplication of a pre-existing operon [42,43].

## Systematic phylogenetic distance-based clustering of EPS operon loci and genomic-proximity networks identifies evolutionarily distinct operon clades

To better understand how these evolutionary events may have altered operon function, we next devised an agnostic, systematic classification strategy to cluster each family of EPS operon loci on the basis of phylogenetic distance (**Fig 2A; see Methods**). In brief, for each EPS operon locus, multiple sequence alignments were generated and used to construct phylogenetic trees. From these trees, we defined sets of clusters through an iterative scan of the tree structure that captures an increasing sequence distance between family members, starting at the leaves and ending at the root. During this scan, sequences are grouped into a cluster if they share a common node (i.e. are within a specified evolutionary distance). To define the optimal set of clusters for each locus, we then applied three cluster quality scoring schemes (Q1, Q2 and Q3) based on the following metrics: proportion of sequences clustered (to maximize the number of sequences clustered); average silhouette score (to minimize the occurrence of clusters containing highly divergent sequences); and the Dunn index (to maximize the separation of closely related sequences from divergent sequences). For each scoring scheme, we defined the optimal pattern of clustering based on the evolutionary distance (expected number of substitutions per site) derived from a maximum-likelihood constructed phylogenetic tree (**see Methods for more detail**s) that maximizes the quality score. Comparisons across scoring schemes (see below) for cellulose operon loci identified Q2 as providing the most informative sets of clusters. Applying this scoring scheme to all EPS loci revealed the average number of sequence clusters generated correlated with the total number of operons predicted for each type of EPS system (**Fig 2B**), which further corresponded to the underlying differences in species

**(A) Overview of Clustering**

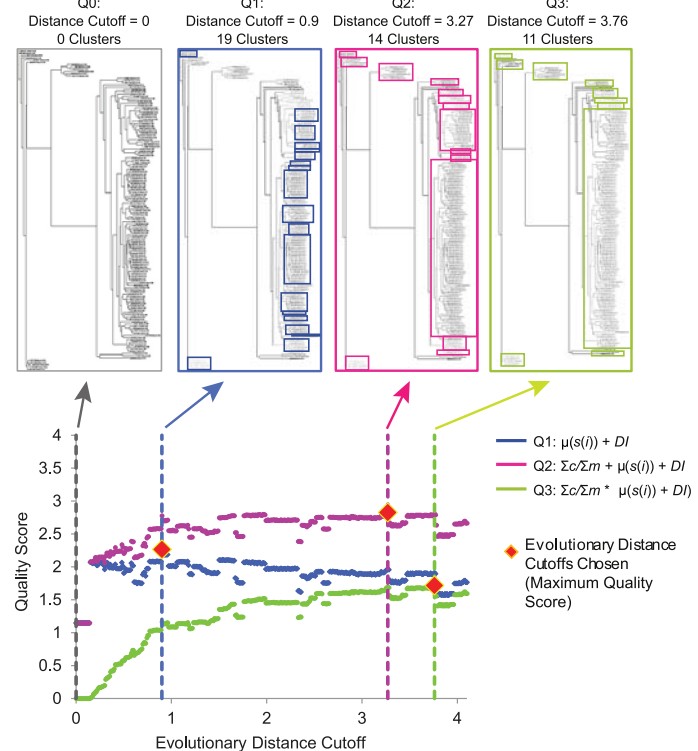

**(B) Predicted Clusters by EPS type**

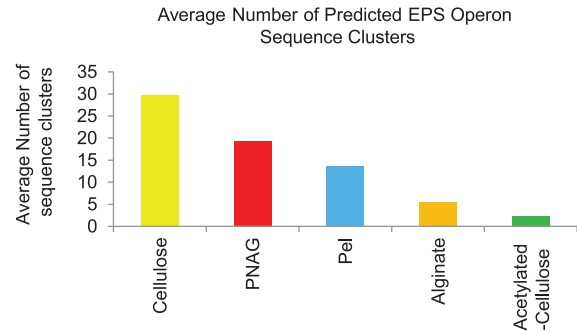

**(C) Cluster Diversity by EPS Locus**

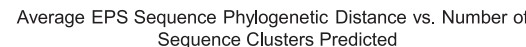
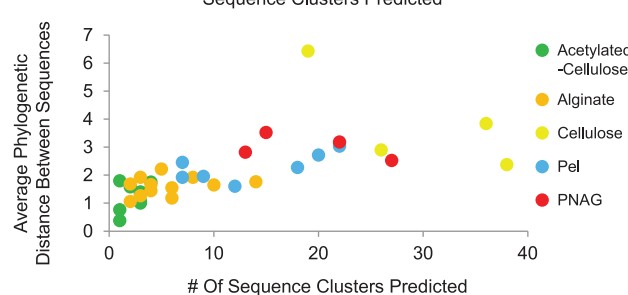

**(D) Operon Networks Generated Using Different Quality Scores**

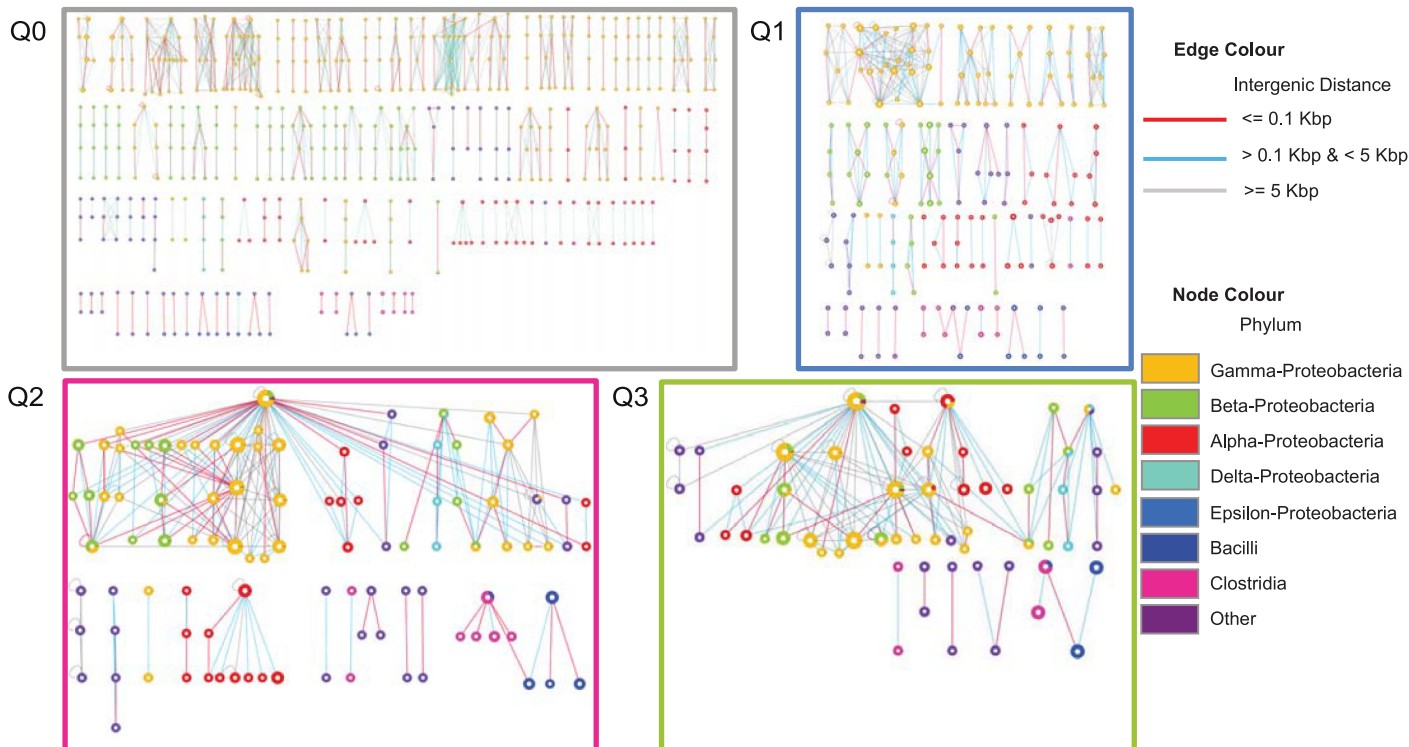

**Fig 2. Clustering of EPS loci.** (A) Schematic illustrating the process of scanning through a phylogentic tree and identifying sets of clusters associated at different evolutionary distance cutoffs. Here evolutionary distance is defined as the number of expected amino-acid substitutions normalized over the multiple sequence alignment length. To identify optimal patterns of clusters, we examined three scoring schemes (Q1, Q2 and Q3). Q1 is defined as the sum of the average silhouette score for all clusters: $\mu(s(i))$ and the Dunn index (DI). Q2 is defined as the sum of the proportion of sequences identified in clusters ($\Sigma c/©m$), $\mu(s(i))$ and DI. Q3 is defined as the product of ($\Sigma c/\Sigma m$) and the sum of $\mu(s(i))$ and DI. For the family of genes related to the *bcsA* locus, each scoring scheme identifies a different optimal evolutionary distance cutoff resulting in defining different sets of clusters. (B) Graph illustrating the average number of sequence clusters predicted (sum of # of clusters over all loci / total number of EPS loci) for each type of EPS operon. (C) Graph illustrating the average evolutionary distance of EPS loci cluster members with other members of the same cluster. (D) Cellulose operon networks generated using the different types of scoring scheme cutoffs used in (A). For each network, nodes indicate clusters of sequences representing individual cellulose loci, edges indicate genome proximity between the two linked loci. Nodes are organized into sets of four, ordered from top to bottom as *bcsA*, *bcsB*, *bcsZ* and *bcsC*. Node size indicates the number of family members associated with that locus cluster. Node colour indicates phylogenetic representation of cluster members. Edge colour indicates genomic proximity of phylogenetic clusters. At higher evolutionary distances (as defined by Q2 and Q3), networks yield more informative patterns of evolutionary relationships as illustrated by larger clusters of loci featuring larger number of interconnections.

https://doi.org/10.1371/journal.pcbi.g002

distributions of EPS systems (**Fig 1D**). For example, the cellulose system was predicted to have the largest average number of sequence clusters overall (30 clusters) and also had the greatest species diversity (Shannon index 2.16 –**S2 Fig**) compared to all other systems. Furthermore, for each EPS system the variability of the number of sequence clusters predicted per locus (**Fig 2C**) suggests differing degrees of locus evolution that are likely to be the result of different structural and functional constraints. For example, a higher degree of conservation would be expected for glycosyl transferase (GT) subunits to maintain efficient co-ordination between polymerization and inner membrane transport of EPS, while increased variability of periplasmic modification enzymes suggests that only a subset of highly conserved motifs are required to carry out polysaccharide modification reactions.

To compare patterns of clusters identified by each scoring scheme, we applied the three scoring schemes to each set of genomically-neighbouring protein sequences assigned by HMM searches cellulose EPS machinery. Here we focused on four core cellulose genes: *bcsA*, *bcsB*, *bcsZ*, and *bcsC* encoding the inner membrane GT, co-polymerase, periplasmic hydrolase and outer membrane export pore, respectively [36]. While previous studies have identified additional genes associated with the production of cellulose biofilms (e.g. *bcsD*, *bcsE*, *bcsF*, *bcsG* and *bcsQ*; [36]), to simplify our analyses we limited our query to *bcsA*, *bcsB*, *bcsC*, and *bcsZ* based on the observation that these are the most abundant genes when comparing all identified cellulose biosynthesis operons. Although this may bias the diversity of operons identified, given that a functional cellulose biosynthesis locus should contain the synthase genes *bcsA* and *bcsB*, we expect our analysis should result in few false negative predictions. From the resulting clusters we generated operon genomic-proximity networks (**Fig 2D**). These networks provide a visual display of the conservation of individual loci, together with their respective genomic proximity to yield patterns of sequence divergence associated with the emergence of distinct forms of operon organization. In the absence of any clustering (Q0), the network trivially resolves into individual operons featuring up to four loci. Applying the Q1 scoring scheme to each locus, the network reveals a variable number of clusters across operon loci, with each cluster generally comprising sequences belonging to the same bacterial genus. Application of the Q2 scoring scheme results in the generation of clusters of increased size, encompassing species featuring distinct operon organizations and compositions. For example, two distinct lineages of alpha-proteobacterial cellulose operons can be easily distinguished, one of which is more closely related in sequence and composition to gamma-proteobacterial operons, and a second which lacks two loci and appears evolutionarily divergent from gamma-proteobacterial operons [25]. However, these distinctions were more difficult to resolve using the Q3 scoring scheme due to clustering of highly divergent sequences. Given the trade-off between clustering highly divergent sequences (Q3) with the depiction of individual operons (Q1), we applied the Q2 scoring scheme to generate clusters for all EPS loci (**S6 Table**).

Using this locus-specific phylogenetic clustering approach, we were able to devise a classification scheme to define EPS locus clades based on the average evolutionary distance of a group of clustered locus sequences to a reference operon sequence (**S3 Table**). For example, the cellulose polysaccharide synthase locus, *bcsA*, from *Escherichia coli* is assigned to clade 1, while divergent alpha-proteobacterial species including *Rhodobacter sphaeroides* are assigned to clade 2. We further resolved operons into distinct groups based on the genomic co-occurrence patterns of locus clades; e.g. for the cellulose operon (*bcsABZC*) we identify clade combinations of 1:1:1:1, 1:2:2:2 and 1:3:5:3, which correspond to operons identified in *Escherichia* spp. and other closely related enterobacteria, *Klebsiella* spp., and *Burkholderia* spp., respectively.

## Phylogenetic clustering and genomic proximity networks reveal evolutionary events driving EPS operon divergence

Having generated clustering patterns for each EPS locus, we next used the sets of cellulose operon associated loci, *bcsA*, *bcsB*, *bcsZ*, and *bcsC*, to examine how these patterns might inform on the evolution of this EPS system. Relative to BcsA, the three other subunits (BcsB, BcsZ and BcsC) display greater sequence diversity as indicated by a larger number of sequence clusters (**Fig 3**). Detailed structure-function studies of the BcsA-BcsB inner membrane cellulose synthase complex, outlined below, illustrate how these findings are consistent with their known functional roles. Further inspection of the cellulose operon network identifies a number of sub-networks comprised of taxon-specific loci clusters associated with distinct patterns of operon organization as illustrated through the following examples: 1) a subnetwork comprised of loci from several beta-proteobacteria, represented here by *Burkholderia cenocepacia* and *Pandoraea promenusa* (**Fig 3(i)**), which feature a rearrangement of the *bcsA* locus and novel locus gains (also supported by inspection of corresponding Genbank genomic annotations) as indicated by a genomic distance of > 0.1 kb between *bcsA* and the neighbouring locus *bcsC*; 2) a subnetwork composed of loci from several species of the alpha-proteobacterial *Zymomonas* show rearrangement of *bcsZ* and/or the loss of *bcsB* or *bcsZ* (**Fig 3(ii)**). Further inspection reveal such losses were due to gene fusion events; 3) a subnetwork composed of loci from a separate group of alpha-proteobacteria which reveals a diverse set of *bcsB* loci that additionally lack the *bcsC* outer membrane pore (**Fig 3(iii)**); 4) a subnetwork of loci from a group of gamma-proteobacteria reveal instances of HGT and divergence (**Fig 4A**). In this latter example, our network identifies two distinct clades of operons, sharing a common group of *bcsA* loci, but featuring two evolutionarily divergent sets of *bcsB*, *bcsZ* and *bcsC* loci which co-occur in several genomes separated by inter-genic distances greater than 10 kb. Detailed investigation of the operonic arrangements of species possessing single copies of either of these clades of operons reveal two distinct loci organizations: the first representing the canonical cellulose locus order (clade A1), *bcsABZC*, found among *E. coli* and *Salmonella enterica* strains; while the second represents a non-canonical locus ordering (clade B1), in which the periplasmic glycoside hydrolase, BcsZ, has undergone a rearrangement, *bcsABCZ*. This clade is found among *Dickeya*, *Erwinia* and *Pantoea* spp. (**Fig 4B**). Of note, we found that several species (e.g. *Enterobacter* and *Klebsiella* spp.) possess both operon clades. The clades have been suggested to have originated by HGT [19]; a hypothesis further supported by our phylogenetic clustering assignments (**Fig 4C**). Furthermore, we identified two additional divergent BcsB sequences associated with a novel organization of operon clade B1 and include several loci with other roles in cellulose production (designated operon clade B2; **Fig 4D**). The divergence of BcsB sequences associated with clade B2 were also found to distinguish bacterial genomes possessing multiple cellulose operons of distinct evolutionary lineages: *Proteus mirabilis* (2 cellulose

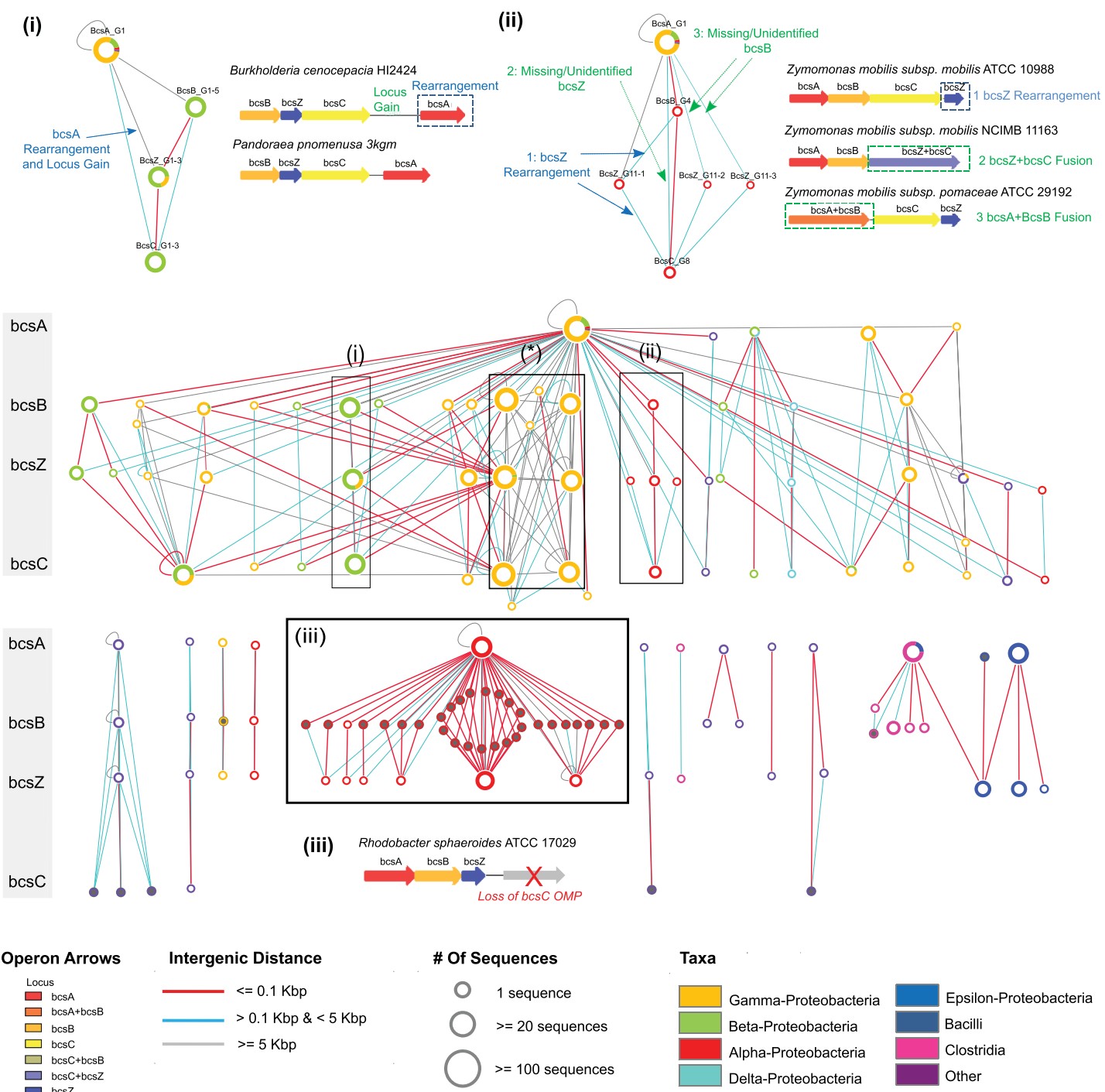

**Fig 3. Genomic-Proximity network of phylogenetically clustered cellulose operons.** Phylogenetically clustered operon loci are arranged vertically with respect to the canonical ordering of the cellulose operon (indicated by grey side bar). Inset boxes depict selected examples of cellulose operon clades, illustrating how the network can inform on evolutionary events: (i) Rearrangement of bcsA among betaproteobacteria–Here, *bcsA* appears in closer proximity to *bcsC* than to *bcsB* or *bcsZ* (as indicated by a cyan coloured edge for the former and a grey coloured edge for the latter). Further the cyan edge indicates a relatively large intergenic distance, suggesting a locus gain between *bcsA* and *bcsC*, confirmed upon inspection of the genome of *Burkholderia cenocepacia*; (ii) Rearrangement and gene fusions in alpha-proteobacteria–in examples 1 and 2, the red edge indicates operons in which *bcsB* is closer to *bcsC* than *bcsZ*, the cyan edges suggest that *bcsZ* is present, but appears after *bcsC* (example 1), while in other operons, *bcsZ* appears missing (example 2). Detailed inspection of example operons (e.g. *Zymomonas* spp.) reveals the fusion of the periplasmic hydrolase and outer membrane pore (BcsZC), in example 3, the apparent loss of *bcsB* in another *Zymomonas* spp. is explained by a fusion between the inner membrane cellulose synthase complex subunits (BcsAB); (iii) Loss of outer membrane pore, BcsC, and divergence of the inner membrane cellulose co-polymerase, BcsB, in alpha-

proteobacteria–in these taxa, BcsB appears highly divergent (as indicated by their identification through more sensitive HMM searches–grey nodes) and no BcsC was identified (confirmed through inspection of representative operons). Further interpretation of the operons identified in the box denoted with an asterisk '*', which represent HGT events, are illustrated in Fig 4. Node size indicates the relative number of sequences per phylogenetic cluster; node colouring represents the taxonomic distribution of loci for a given cluster; edges connect clusters which co-occur in the same genome(s); edge colour indicates the genomic-proximity of loci clusters.

operons: Clades A1 and B2) and *Enterobacter* spp. (3 cellulose operons: Clades A1, B1 and B3) (**Fig 4E**). Additional sequence database searches revealed that the non-core loci associated with operon clades B2 and B3 share functionally homologous loci to the cellulose accessory protein D (AxcesD), which has been characterized as increasing the efficiency of cellulose production in the *Acetobacter xylinus* cellulose synthase complex [44]; GalU an uridine triphosphate (UTP)-glucose-1-phosphate uridylyltransferase involved in cellulose precursor biosynthesis; and an additional uncharacterized locus predicted to possess both PAS_9 and GGDEF signaling domains, indicating the potential adaptation in *Proteus* and *Enterobacter* spp. to produce varied forms of cellulose upon different environmental stimuli [45].

### Genomic-proximity networks of pel operons reveal a novel *pel* locus in the gram positive bacterium, *bacillus cereus* that is regulated by c-di-GMP

Examination of the genomic-proximity networks of *pel* loci also reveal novel operon organizations across phylogenetically divergent bacteria (**Fig 5**). As with cellulose loci *bcsA* and *bcsZ*, we identify examples of operon rearrangements involving *pelB* (outer membrane transport pore and TPR domain) loci and *pelA* (periplasmic modification hydrolase) (**Fig 5(ii), 5(iiib) and 5(iv)**), across several species associated with diverse environments. Again, consistent with our findings for cellulose, we noted loci losses and acquisitions. Although it has not been demonstrated that the *pel* operon forms a trans-envelope biosynthetic complex, the ordering of operon loci has been shown to play an important role in the assembly of macromolecular complexes [46] and optimizing biosynthetic pathways [47], suggesting that there exists a functional coupling between Pel outer membrane transport and periplasmic modification [48]. However, the effects of these rearrangement events on Pel production still remain to be experimentally investigated.

We also observed a high degree of overall conservation among components which are known to play key roles in Pel biogenesis, such as the putative polysaccharide synthase (PelF), putative inner membrane protein (PelG), hydrolase/deacetylase (PelA) and cyclic-di-GMP receptor (PelD) [32]. In contrast, a greater degree of divergence can be seen among inner (PelE) and outer membrane (PelB, PelC) transport associated loci. In these loci, there appears a consistent pattern of clustering across bacterial phyla suggesting co-evolution of potentially physically interacting components, however no evidence of interaction has been shown to date.

Our genomic proximity network revealed two distinct clades in several Gram-positive species (**Fig 5(v)**). Of the synthase dependent EPS operons known to date, only PNAG production has been genetically and structurally characterized in Gram-positive Staphylococci [49]. Operons reconstructed from initial HMM searches identified putative *pel* operons in several Gram-positive bacteria, comprised of the GT encoding PelF and the PelG putative transport protein (**Fig 5**). To determine whether these were bona-fide *pel* operons with additional loci, iterative HMM searches were performed including additional protein sequences from predicted *pel* operons. These searches revealed additional loci including a homolog of PelD (**S3 Fig**). C-di-GMP signaling in Gram-positive bacteria is less well characterized [50] and this finding suggests a role for this secondary metabolite in regulating biofilm formation in these species. In our companion paper, we have experimentally validated our predictions by showing that single gene deletions within the predicted *B. cereus pel* operon result in a loss of EPS production, and that PelD regulates EPS production through binding of c-di-GMP (Whitfield *et al* PLoS

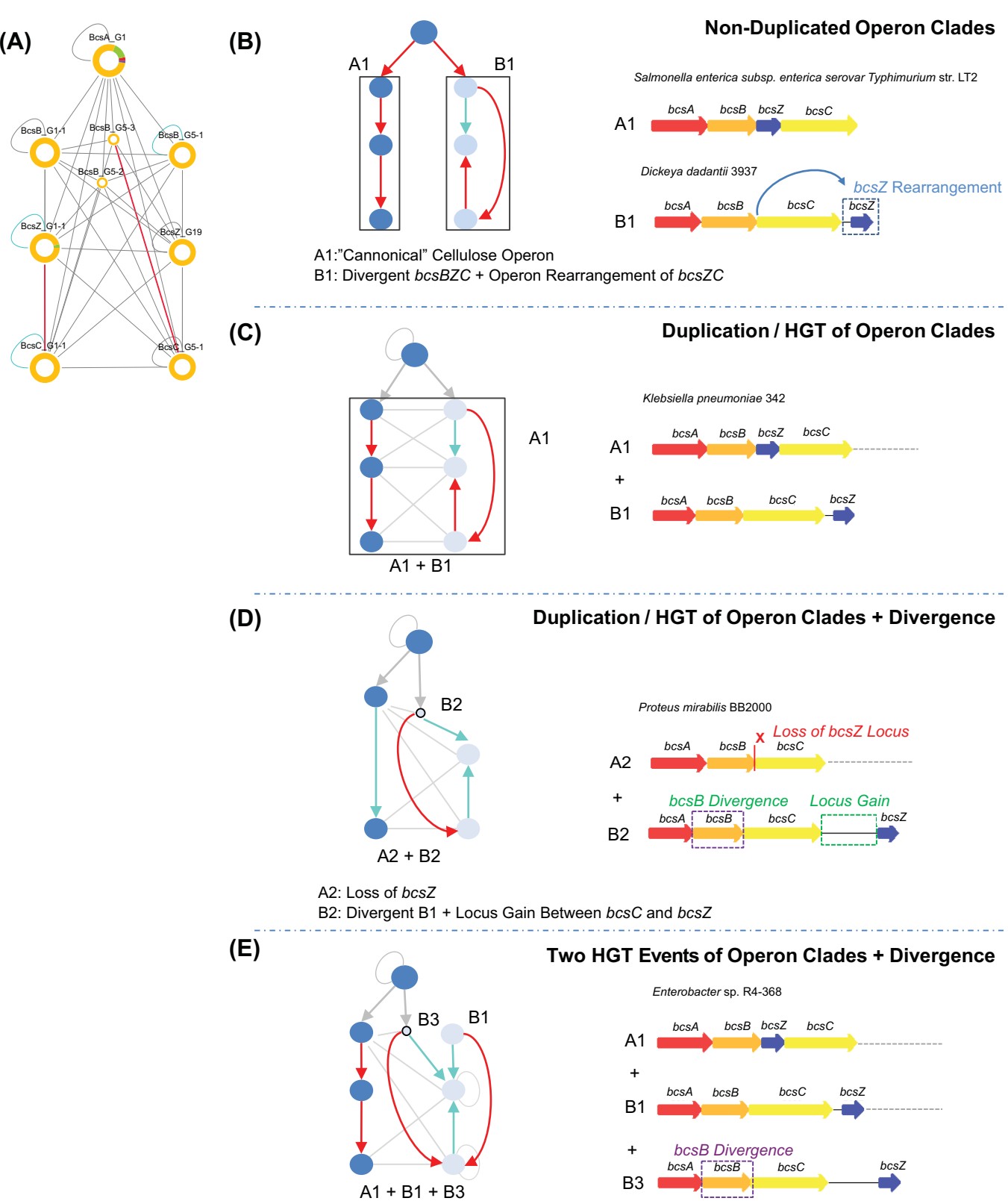

**Fig 4. Horizontal gene transfer of cellulose operons identified from analysis of the genomic-proximity network.** Here we show how a subgraph (A) from the global cellulose EPS operon genomic-proximity network (**Fig 3**(*)), may be interpreted to reveal HGT events involving two distinct gamma proteobacterial operon clades, A (canonical *bcsABZC*) and B (*bcsABC-Z*). (B) Examples of operons in two species which possess either a single A1 ("canonical") or B1 (rearrangement of *bcsZC*) operon clade. (C) Example from *Klebsiella pneumoniae* in which a single genome contains both A1 and B1 operons, indicating a HGT event. (D) Example from *Proteus mirabilis* featuring two copies (designated A2 and B2 respectively) of the cellulose EPS operon, which appear to be divergent forms of A1 and B1: A2 features an apparent loss of the *bcsZ* locus from A1; B2 features a locus gain between *bcsC* and *bcsZ* from B1. Example from *Enterobacter* spp. in which the genome carries three copies of the cellulose EPS operon. In addition to clade A1 and B1 operon arrangements, a further operon (designated B3) appears in which *bcsB* has diverged from a B2 clade operon. Arrows within the network schematics depict the order of loci within the operon and are coloured according to intergenic distance: red < 100bp; cyan >100bp & <5 kb; grey >5 kb.

Pathogens, in press). This work provides the first and crucial piece of evidence which suggests that divergent Gram-positive *pel* operons, particularly those belonging to the same phylogenetic clade as *B. cereus*, likely possess the ability to produce a Pel-dependent biofilm. However, it is also possible that significantly divergent operon loci may not result in the production of EPS of identical composition to that characterized in Gram-negative bacteria, as in the case of PNAG modification between Gram-negative (*pga*) and Gram-positive (*ica*) operons [51].

## Genomic-proximity networks of PNAG uncover locus loss and duplication events in pathogenic and environmental bacteria

To examine how locus duplication, loss, and rearrangement events have contributed to the evolution of PNAG operons across bacterial phyla, selected examples of *pga* operon clusters were identified and compared (**S4 Fig**). For example, within a group of enterobacteria possessing related *pgaD* loci, there exist a number of closely related pathogenic enterobacteria that have lost *pgaA* (*E. coli* ETEC H10407), as well as *pgaB* (*Shigella flexneri* 5 str. 8401). These losses suggest these taxa may no longer be able to produce PNAG (**S4(i.a) Fig**). Interestingly, we also observed a lack of *pga* operons among pathogenic *Salmonella* spp. genomes surveyed in this study. Previous work has shown that loss of PNAG production is associated with adaptation of *Salmonella* spp. to an intracellular pathogenic lifestyle [52]. Although PNAG production in *E. coli* H10407 has not been examined, our findings are also consistent with the adaptation of *E. coli* H10407 from a commensal to a pathogenic lifestyle [53], where enterotoxins and colonization factors serve a crucial role for attachment to the intestinal epithelium and enhanced toxicity [54,55]. Furthermore, it has been previously shown that biofilm formation in *S. flexneri* impairs invasive ability and virulence [56], which suggests that the loss of PNAG production in *S. flexneri* [57] is the result of adaptation to an intracellular mode of infection. These results shed light on the relationship between biofilm production capability and adaptation of enteric bacteria toward a pathogenic lifestyle.

Based on the divergence of *pgaB* loci, we also identified *pga* operon clades corresponding to partial and whole operon duplications in aquatic bacteria, including a partial duplication of the *pga* operon specific to the important pathogen *Acinetobacter baumannii* spp. and *Methylovora versatilis* 301, respectively (**S4(ii) Fig**). Also, in environmental bacteria we discovered a novel *pga* organization resulting from rearrangement of *pgaC*, and a lack of *pgaB* and *pgaD* loci, which may have been too divergent to detect from initial HMM searches (**S4(iii) Fig**). Although our HMM models only used Gram-negative *pga* operon protein sequences, we also identified a number of Gram-positive *pga* operons consisting of *pgaB* and *pgaC* (**S4(i.b) and S4(i.c) Fig**). Upon closer inspection these loci were found to correspond to *Staphylococcus* polysaccharide intercellular adhesion (PIA) loci *icaB* and *icaA*, respectively. This suggests a potential common evolutionary origin of synthase-dependent PNAG production between Gram-positive and -negative organisms.

A clade of *pga* operons were also identified possessing varying numbers of divergent *pgaC* loci resulting from repeated tandem duplication events (**S4(v) Fig**). Despite lacking a

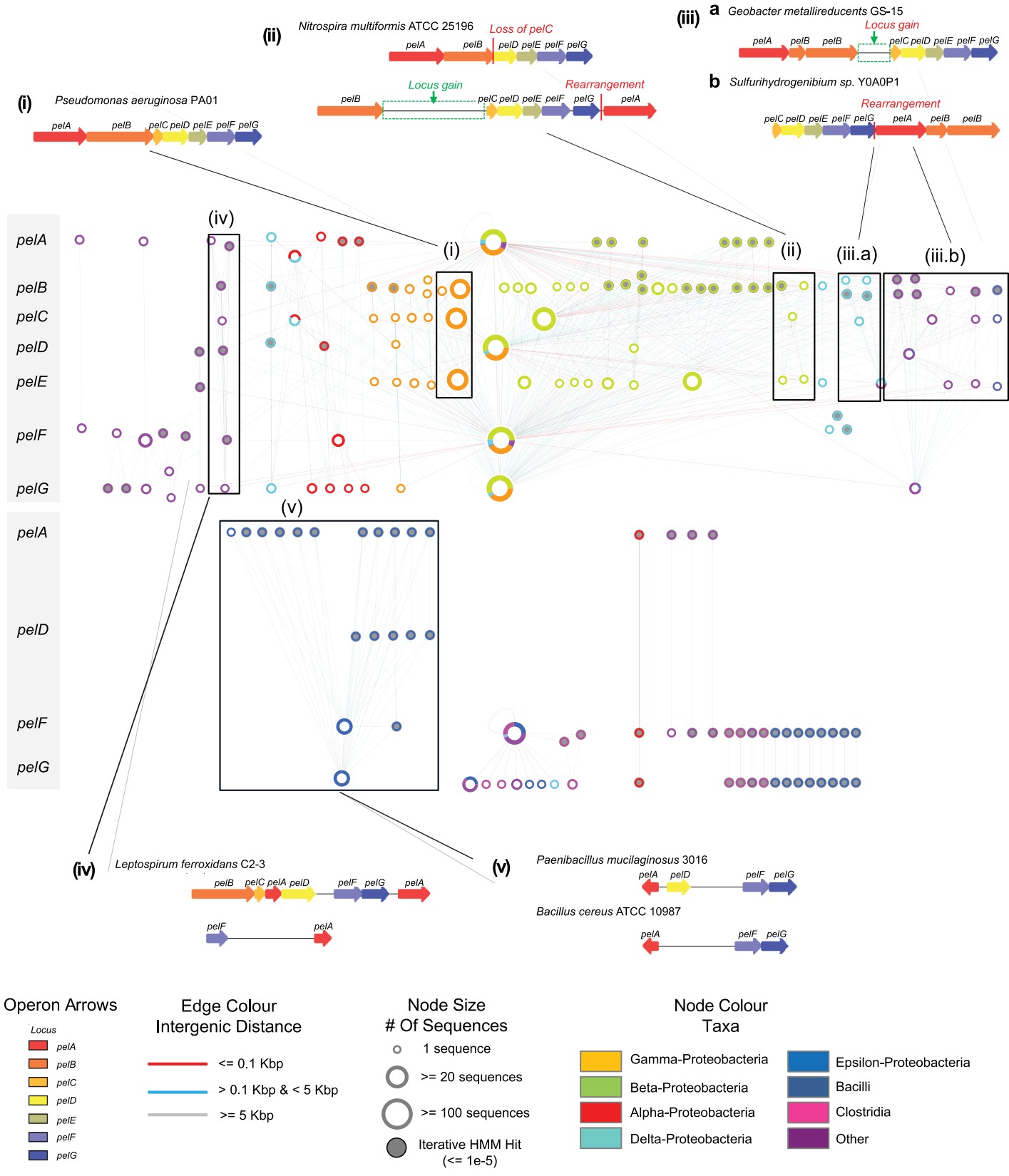

**Fig 5. Genomic-proximity network of phylogenetically clustered *pel* operons.** Phylogenetically clustered operon loci are arranged vertically with respect to the canonical ordering of the *pel* operon (indicated by grey side bar). As for Fig 4, inset boxes depict selected examples of *pel* operon clades, illustrating how the network can inform on evolutionary events: (i) Canonical organization of the *pel* operon, as defined in the *Pseudomonas aeruginosa* genome.; (ii) Duplication of the pel operon in *Nitrosospira multiformis* with subsequent evolution through locus gain and loss, as well as rearrangement of *pelA*; (iii) *pelB* fission, locus gain and rearrangement in aquatic thermophilic species; (iv) A potentially novel duplicated *pel* operon identified in *Leptospirillum ferrooxidans* comprised of divergent *pelA* and *pelF* loci; (v) *pel* operons identified in Gram-positive species including divergent *pelD* loci involved in regulation through c-di-GMP. Node size indicates the relative number of sequences per phylogenetic cluster; node border colouring represents the taxonomic distribution of loci for a given cluster; grey filled nodes indicate loci predicted by iterative HMM searches; edges connect clusters which co-occur in the same genome(s); edge colour indicates the genomic-proximity of loci clusters.

detectable *pgaA* locus, a possible role of these gene clusters in EPS production was investigated. One member of this operon clade, *Thauera* sp. MZ1T, inhabits a wide range of environments, and is an abundant producer of EPS responsible for viscous bulking in activated sludge waste-water treatment processes [58]. Furthermore, a recent mutagenesis study [59] demonstrated that biofilm-formation defective *Thaurea* mutants could be rescued by the complementation of the predicted *pgaB* deacetylase locus identified in the present study. Together, these findings suggest that the divergence of deacetylase and duplication of GT related loci in PNAG biosynthesis have resulted in the emergence of a distinct operon lineage.

### Genomic proximity networks of alginate uncover distinct operon clades in *pseudomonas* spp. and atypical operon architectures in environmental bacteria

Although the majority of alginate operons were predicted largely among *Pseudomonas* spp. genomes (**S5 Fig**), phylogenetic clustering and genomic-proximity network reconstruction revealed an array of events influencing alginate operon evolution. For example, two distinct alginate operon clades were identified among *Pseudomonas* spp., defined by whole operon duplication and rearrangement of alginate polysaccharide modification loci (**S5(i) and S5(ii) Fig**). Also identified were divergent, "atypical", alginate operons (**S5(iii) Fig**) comprising extensive rearrangements and also losses of functionally related subsets of alginate loci, *e.g.* outer membrane transport loci (*algKE*), and polysaccharide modification machinery (*algGX-LIJF*). Closer examination of the alginate genomic-proximity network also indicated a greater number of clusters for *alg44* and *algX* loci, which were reflective of increased divergence among distinct alginate operon clades. Given that both loci play related roles in the regulation, polymer-modification, and assembly of the alginate EPS secretion machinery [60], these results provide an avenue for future research toward elucidating how species may modify alginate production to adapt to diverse environmental niches.

### Genomic proximity networks of acetylated cellulose operons reveals duplication of co-polymerase subunits and sequence homology of loci with alginate acetylation machinery

From the genome sequences surveyed, only four species were identified as possessing acetylated cellulose operons, comprising two distinct operon clusters with differing operon constitutions among three *Pseudomonas* spp. and *Bordetella avium* 197N (**S6 Fig**). Contrary to cellulose phylogenetic clusters, the polysaccharide synthase, *wssB*, was divided into distinct Gamma- and Beta- proteobacterial clusters. We also found a distinct phylogenetic cluster identifying a unique tandem duplication of *wssC* in *Bordetella avium* 197N, which was not observed among orthologous cellulose *bcsB* co-polymerase loci (**S6(ii) Fig**). This observation might suggest a divergent mechanism of action of cellulose inner membrane transport. As we previously observed (**Fig 1E**), 3 out of 4 of the predicted acetylated cellulose operons were also found to co-occur with alginate operons. Additional HMM-searches identified significant

sequence similarity between acetylated cellulose *wssBCDE* operon sequences to those previously identified as *bcsABZC*, as well as between acetylated cellulose acetylation-machinery and their functional homologs in alginate operons (WssH–AlgI; WssI–AlgJ/AlgX). Taken together, these findings suggest that acetylated cellulose production has likely evolved through the duplication and operonic acquisition of the alginate acetylation machinery loci.

## Sequence variability of phylogenetic clusters reveals different degrees of structural conservation of cellulose biosynthesis machinery

With the availability of a crystal structure for the BcsA-BcsB inner membrane complex responsible for cellulose biosynthesis and transport [61], we examined the potential structural and functional consequences of the sequence variability of the BcsA and BcsB phylogenetic clusters highlighted above (**Fig 3**). In brief, we generated multiple sequence alignments of eight BcsA and BcsB sequences summarizing the evolutionary diversity of cellulose operon clades identified in **Fig 3**. Residue conservation information from this alignment were subsequently mapped onto the structure of the BcsA-BcsB complex (PDB ID:4HG6 [62]; **S7 Fig**). The results of the following analysis are also consistent when including all predicted BcsA and BcsB sequences. We identified a high degree of sequence conservation among BcsA sequences corresponding to the GT domain responsible for cellulose polymerization. Conserved residues mapped specifically to a cleft in the GT domain where a uridine diphosphate (UDP) carrier moiety is bound and oriented through a conserved QxxRW motif to enable polymerization of glucose monomers of the growing cellulose chain [61]. Conversely, the PilZ domain of BcsA, involved in regulation of the GT function in response to c-di-GMP levels shows low conservation overall, except for the subset of residues required for c-di-GMP binding. Further, the periplasmic region of BcsB shows low sequence conservation overall, aside from a number of highly conserved residues in the carbohydrate binding and ferredoxin domains. These residues include one (L193 of the *Rhodobacter sphaeroides* ATCC 17025 reference sequence) representing a putative cellulose binding residue oriented in close proximity to the growing cellulose chain near the exit of the BcsA inner membrane translocation channel. From phylogenetic sequence clustering, structurally relevant conservation features of the cellulose synthase complex were identified which should facilitate further investigation of cellulose EPS production across phylogenetically diverse species. For example, c-di-GMP binding residues of the PilZ domain of BcsA vary in conservation across phylogenetic clusters, which could impact the ability of the protein to bind the nucleotide. This may in turn limit access of activated glucose monomers to the GT domain, thus altering the rate of cellulose polymerization. Insertion/deletion events are also observed across BcsB phylogenetic clusters that may facilitate the recruitment of additional periplasmic processing proteins [63], or macromolecular assembly of the BcsA-BcsB complex [64]. This might result in differences in the higher-ordered structuring of cellulose microfibres as a consequence of adaptation to diverse environmental niches. These results demonstrate how the application of our phylogenetic clustering methodology can be extended to provide biologically informative insights into the function of components of EPS secretion machineries.

## Phylogenetic clustering elucidates the structural and functional divergence of the *pgaB* locus, revealing the evolution of PNAG production across gram-negative and gram-positive bacteria

PNAG production is found across phylogenetically diverse species and is carried out by the *pgaABCD* operon of Gram-negative bacteria [39] and *icaADBC* operon of Gram-positive bacteria [65]. Although the functional and immunological properties of PNAG produced by the

*pga* and *ica* loci appear to be similar [66], there are important differences between the roles of *pga* and *ica* operon loci [67]. Common to both operons is the presence of an integral membrane GT locus, *pgaC* and *icaA*, which are both members of the GT-2 family and share sequence homology [67]. In addition, non-homologous loci encoding integral membrane proteins, *pgaD* and *icaD*, are also present and required for the full function of their respective GTs [66,68]. In Gram-negative bacteria, PNAG production is regulated through physical interactions between PgaD and PgaC which are stabilized by the allosteric binding of c-di-GMP [56]. In Staphylococci, PNAG production does not depend on c-di-GMP and is likely regulated by an alternate signaling pathway [69]. Deacetylation of PNAG is carried out by *pgaB* and *icaB* loci and has been shown to play a crucial role in biofilm formation and immune evasion [51,70]. *pgaB* also possesses an additional C-terminal glycoside hydrolase domain which cleaves the PNAG polymer following its partial deacetylation [71], although the mechanism of how these activities are coordinated and the biological role of the hydrolase activity is unknown. Unique to *pga* operons is a locus encoding an outer membrane export pore, *pgaA* [72] and, in *ica* operons, an additional integral membrane protein, *icaC*, which has been proposed to be involved in PNAG O-succinylation [67]. Using Gram-negative *pga* loci as seed sequences for the reconstruction of synthase-dependent PNAG operons, we were also able to identify Gram-positive *ica* operons based on significant sequence similarities to *pgaB* and *pgaC* loci. Our phylogenetic clustering approach also revealed that *pgaC/icaA* sequences clustered into a single clade, while *pgaB/icaB* were associated with distinct sequence clades (**S4 Fig**). To explore the evolution of Gram-negative and Gram-positive *pga* and *ica* operons, we generated multiple sequence alignments for representative sequences of 18 PgaB clades. Our phylogenetic clustering results confirm previous observations [67] that the glycoside hydrolase domain is exclusively associated with Gram-negative *pga* operons (PgaB_G1) and is absent in a clade of *Staphylococcus* Gram-positive *ica* sequences (PgaB_G3; **S8A Fig**). We also identified additional Gram-positive *icaB* clades among non-*Staphylococcus* spp., e.g. *Bacillus* and *Lactococcus* (**S1 Table**), which possess operons lacking the *icaC* locus [67]. Interestingly, we also identified a number of divergent Gram-negative *pgaB* clades resembling *icaB* clade sequences. Members of these clades lacked the canonical N-terminal glycosyl hydrolase domain, and were distinguished by possessing N-terminal fusions, primarily of GT domains. Furthermore, these *pgaB* clades are associated with operons lacking detectable *pgaA* outer membrane pore locus and *pgaD* (**S4(v) Fig**). Although PNAG production in these species has not been experimentally confirmed, these findings suggest that if the polymer is produced, it may be regulated through a novel mechanism, that glycoside hydrolase activity might not be essential for PNAG export across all Gram-negative species, and that other modes of export may exist. The N-terminal fusion of GT with PgaB de-acetylase domains would also suggest that the de-acetylase activity of PgaB in these organisms may be associated with the periplasmic face of the inner membrane, in contrast to dual domain PgaB clades where the protein is predicted to function at the periplasmic face of the outer membrane [72].

In addition to these novel domain fusion events, PgaB phylogenetic clustering enabled us to resolve distinct events affecting the evolution of the deacetylase domain across different operon clades. Using the *E. coli* K12 MG1655 sequence of the largest PgaB clade (PgaB_G1) as a reference, multiple sequence alignments against other representative PgaB clade sequences identified several regions of insertion/deletion events (**S8A Fig**). When these regions were mapped to the published crystal structure of PgaB (PDB ID: 4F9D [73]), they were found to correspond to distinct structural elements surrounding the conserved deacetylase core (**S8B and S8C Fig**). We assigned insertion/deletion regions a number according to their order of appearance in the multiple sequence alignment of PgaB deacetylase domains, and divided them into two categories (**S8D Fig**). The first two indel regions, 1 and 2, resided in the N-

terminal region of the reference *E. coli* sequence, and corresponded to beta-strands flanking the conserved active site residues involved in deacetylation, His55, Asp114, and Asp115. Region 1 was associated with Gram-positive *icaB* and comprised insertions of ~10aa in *Staphylococcus aureus* VC40 (PgaB_G3), as well as *Bacillus infantis* NRL B-14911 (PgaB_G7), *Lactobacillus plantarum* 16 (PgaB_G9), *Leptospirillum ferriphilum* ML-04 (PgaB_11). Structural characterization of *Ammonifex degensii* IcaB (PgaB_G3) identified residues overlapping with Region 1 as encoding a hydrophobic loop responsible for membrane localization in this species [74]. Region 2 was found to be exclusive to Gram-negative *pgaB* loci and comprised a much larger insert of ~77aa in *Geobacter metallireducens* GS-15 (PgaB_G2), *Crinalium epipsammum* PCC 9333 (PgaB_G5), and *Colwellia psychrerythraea* 34H (PgaB_6). The functional role of this insertion is unknown.

The last three insertion/deletion regions, 3–5, occurred in a region oriented away from the deacetylase active site, and correspond to two beta-turn motifs and an alpha-helix cap, respectively. To further elucidate the biological import of identified PgaB indel regions, we examined regions 3, and 5 in the context of Gram-negative PNAG modification. In the *E. coli* K12 MG1655 PgaB_G1 sequence, region 3 encompasses a beta-turn with an elongated loop, which is spatially proximal to a disordered loop and alpha helix (pos. 367–392) on the N-terminal region of the PgaB glycoside hydrolase domain. Region 3 also encodes a histidine (*E. coli* PgaB —H189) which is part of the nickel binding pocket of Gram-negative PgaB deacetylases. Both regions contain polar and electrostatically charged residues which are highly conserved across PgaB_G1 sequences (**S8E Fig**). Region 5 corresponds to an 8 amino acid elongation of an alpha-helix (pos. 219–226), which also appears to provide an additional point of contact between the deacetylase and hydrolase domains. Although region 5 is also shared with *icaB* associated sequences (PgaB_G3), region 3 appears only in other dual deacetylase-hydrolase Gram-positive *pgaB* sequences identified in the sporulating bacteria *Lachnoclostridium phytofermentans* ISDg and *Kitasatospora setae* KM-5043. Although initial PFAM searches failed to identify the additional Gram-positive C-terminal domains, subsequent BLAST searches revealed them to be homologous to glycoside hydrolases. In region 4 a unique 29 amino acid insertion was also identified in *Lachnoclostridium phytofermentans* ISDg (PgaB_G16), which may play a compensatory role for the absence of 9aa in region 3. These insertion regions suggest an overall functional importance in ensuring stability between each domain and could play a role in coordinating their activities. These findings in combination with our identification of *ica*-like operon organizations among environmental Gram-negative species (**S4(v) Fig**) suggest that Gram-negative *pga* operons may share a common evolutionary origin with Gram-positive *ica* operons. Recent research is providing growing evidence for the emergence of the di-derm Gram-negative architecture from sporulating monodermal Gram-positives [75], which provides a plausible evolutionary context for the insertion/deletion events observed among *pgaB/icaB* deacetylase domains. Through the loss of inner membrane localization [74] (Region 1), the compensatory gain of an N-terminal palmitoylation site [66], along with a C-terminal fusion of a hydrolase domain (Regions 3–5), an ancestral deacetylase locus may have been adapted to regulate the export of PNAG [66] at the outer-membrane of Gram-negative *pga* operon lineage.

## Discussion

In this work we describe a novel and generalizable approach for the systematic classification and presentation of bacterial protein families in the context of their host operon. Protein families are defined as sets of homologs (groups of related sequences having a common evolutionary ancestor) sharing a particular set of sequence motifs or structural domains that can be

utilized to determine their biological roles. For example, the PFAM database utilizes curated sets of protein family sequences in the generation of profile HMMs [76]. A key challenge that complicates the definition of these relationships are evolutionary events such as duplication, gene fusion, and HGT. In attempts to account for such events, a variety of computational approaches have been developed for refining functional assignments. These operate either by graphical clustering of pair-wise protein sequence similarities (e,g, COG [27], OrthoMCL [19] and EggNOG [20]), or through the generation of hierarchical evolutionary relationships and construction of phylogenetic trees (e.g. TreeFAM [77] and TreeCL [78]). However, these methods are limited in their ability to provide further resolution of sequence diversity within a family that might otherwise offer additional insights into evolutionary events that allow taxa to adapt to specific environments.

Agnostic approaches to define sub-clusters of evolutionarily related protein families have ranged from phylogenetic tree reconstructions [79] to hierarchical clustering of pairwise global sequence alignments [80]. Here we present an extension of previous efforts and introduce a novel systematic approach for defining protein sub-family relationships through the clustering of phylogenetic trees. Key to this approach is defining a scoring function that allows a phylogenetic tree to be resolved into optimal clusters that best capture the similarities between cluster members, as well as the dissimilarities between clusters. Combining two clustering quality metrics (Silhouette and Dunn index) and proportion of sequences clustered, we demonstrate that our approach classifies a diverse array of operon-associated protein families into taxonomically consistent and functionally informative sub-clusters. Genomic-proximity networks were also constructed to provide an intuitive means of utilizing phylogenetic clusters to examine diverse mechanisms of operon evolution across taxonomically diverse bacterial genomes. Genomic-proximity networks have previously been utilized for inferring functional relationships [81], understanding mechanisms underlying bacterial genomic organization into functionally related gene clusters [82], and transcriptional regulation of bacterial operons [83]. In this study we extend the application of genomic-proximity networks as a tool for the systematic exploration of operon evolution resulting from locus divergence, loss, duplication, and rearrangement events.

To demonstrate the effectiveness of our approach, we applied our methods to classify the synthase-dependent bacterial EPS operon machineries for 5 different polymers: cellulose, acetylated cellulose, alginate, Pel and PNAG. There has been only one previous attempt to classify synthase-dependent EPS operons and this focused specifically on the cellulose system [36]. In that study, cellulose operons were categorized into four major types, based on the presence or absence of experimentally validated accessory loci involved in cellulose production. Here, we based our analysis on the four core operon loci, *bcsABZC*, deemed essential for cellulose production. Cellulose operon clades identified in this study showed little consistency with the previously defined four major cellulose operon types [36], suggesting that the conservation of accessory loci is more variable across bacterial species compared to loci encoding core EPS functionalities. However, one operon type was identified in this analysis, representing the loss of the BcsC outer membrane transporter identified among a subset of alpha-proteobacterial genomes, which include several known cellulose producing species [62,84] suggesting a novel mechanism of cellulose export (**Fig 3(iii)**) [36]. We also found that the loss of BcsC has resulted in an increased divergence of BcsB loci in these genomes, which highlights the key role of BcsB as an intermediary between cellulose biogenesis and periplasmic transport (**S7 Fig**).

In general, inner membrane components involved in EPS polymerization were found to be relatively conserved across all systems examined, while periplasmic and outer membrane components showed a relatively increased degree of evolution, which are likely to have important functional implications. For example, in the cellulose and Pel operon networks (**Figs 3 and 5 and S5 Table**), rearrangement events involving the periplasmic glycosyl hydrolase (BcsZ) and

glycosyl hydrolase/deacetylase (PelA) were found to be a defining feature of several operon clades. It is interesting to note that these rearrangements have resulted in a change in the ordering of *bcsZ* and *pelA* relative to their respective outer membrane transport pore loci, which highlights the important role of polysaccharide modification in both the biogenesis and regulation of extracelluar EPS transport [48,85,86]. Similarly, the rearrangement of alginate modification machinery loci (*algIJF*) was observed as a distinguishing feature of *Pseudomonas* spp. operon clades. These findings suggest that rearrangement and locus ordering may serve as an important means of regulating EPS production by modifying the timing of translation of modification enzymes, which could affect the assembly of EPS complexes or the structural properties of EPS produced [47,64,87].

Furthermore, identifying operon clades through a phylogenetic approach elucidated numerous instances of cellulose whole operon duplications arising from HGT of two evolutionary distinct operon clades (**Fig 4**). Such large-scale duplications, if they are functional, may either serve as a dosage response to given environmental stressors, as observed in the duplication of bacterial multiple-drug transporter operons [88], or could be under the regulation of different environmental stimuli. Interestingly, representative species of the two cellulose operon lineages identified in HGT events, e.g. the plant and human pathogens, *D. dadantii* and *S. enterica*, respectively, are known to produce structurally distinct forms of cellulose with different properties and roles in pathogenesis [89,90]. In addition, we identified that BcsB divergence was also seen to accompany the rearrangement or horizontal transfer of these operons, which further suggests that it may play a key role in the fine-tuning of cellulose production by coordinating the export of growing cellulose polymers through the periplasm. Furthermore, our analyses of acetylated cellulose, alginate and PNAG operons suggest a dynamic evolutionary scenario for the evolution of EPS biofilm production through the acquisition of novel polysaccharide modification loci. The limited number of acetylated cellulose operons identified, their frequent co-occurrence in alginate possessing species, and significant sequence similarities between acetylation machinery loci, suggests that the cellulose acetylation machinery is likely to have originated from previously existing alginate operons in *Pseudomonas* spp. The evolutionary trajectories of Gram-positive and Gram-negative PNAG operon lineages appears to have resulted through the fusion of glycosyl hydrolase and deacetylase domains in Gram-negative *pgaB* loci.

A further key finding from this study was the identification of homologous *pel* operons in the genomes of several Gram-positive bacteria. With the additional identification of homologs of PelD through iterative HMM searches, our analyses have uncovered a novel example of c-di-GMP regulation of biofilm machinery in Gram-positive bacteria. In the accompanying paper we experimentally validate that a predicted *pel*-like operon in *B. cereus* ATCC 10987 is responsible for biofilm production and is regulated by the binding of c-di-GMP to PelD (Whitfield *et al* PLoS Pathogens, in press).

Together this work demonstrates a novel integrative approach combining phylogenomics and genomic-context approaches to systematically explore the adaptive implications of sequence divergence of protein families associated with operon associated EPS secretion machineries. Further extension of this work holds great potential as a general approach for elucidating how bacterial operon encoded biological pathways and complexes have contributed to bacterial adaptation to and survival in diverse environmental niches and lifestyles.

## Methods

### Sources of data

Sequences corresponding to experimentally characterized EPS operon loci were obtained from the National Centre for Biotechnology Information (NCBI) reference sequence database [91]

(**S3 Table**). Fully sequenced genomes and associated protein sequences were obtained for 1758 bacteria from the NCBI (Retrieved April 20[th] 2015) (**S7 Table**). In this set of genomes the major phyla represented are largely Gram-negative Proteobacteria (47%), followed by Gram-positive Firmicutes (~20%) and Actinobacteria (~10%). For each bacterial strain predicted to possess an EPS operon, metadata corresponding to niche (host-associated or environmental) and lifestyle (pathogenic or non-pathogenic) were collated from literature searches (**S8 Table**– and is also made available for download from https://github.com/ParkinsonLab/eps_biofilms).

### Prediction of EPS operons

To identify putative EPS operons, we applied an iterative HMM-based sequence similarity profiling strategy. For each set of EPS loci, we first constructed a HMM from previously characterized EPS producing bacteria (**S3 Table**); alignments were constructed using MUSCLE v.3.8.1551 [92], with default settings, from which HMM-models were built using HMMER v.3.1b2 [93], with default settings. As the number of characterized EPS producing species varies greatly by system, each HMM was then used to identify additional EPS loci within the set of 1733 bacterial genomes and build an expanded HMM model for downstream operon prediction. First a protein-BLAST search of reference sequence EPS protein coding sequences was performed against the set of reference+representative completely sequenced bacterial genomes downloaded from NCBI (e-value threshold of < = 1e-5). Next, these putative EPS sequences were subject to a second-round of all-vs-all protein-BLAST searches. These results were then processed using an in-house custom perl script to select non-redundant sequences (percentage-identity of < 97%) in an incremental fashion: starting with the reference EPS sequence used in the first step, its highest scoring non-redundant match was selected and subsequently used to identify the next non-redundant sequence with the highest scoring match, and repeated until a pre-defined number of sequences were selected. To capture a consistent degree of locus sequence diversity and HMM sensitivity for each EPS system, we selected 20 sequences to represent each locus with the expectation that this would provide an adequate lower-bound on sampling potential amino acid substitutions for sites undergoing random mutation, i.e. not-under functional constraints. Using these new sets of HMMs, sets of EPS loci for the reconstruction of EPS operons (see below) were predicted through sequence similarity searches of the 1733 genomes using HMMER, with default settings. Significant sequence matches were defined as those with E-values < = 1e-5. HMM models as well as the table detailing lifestyles and environmental niches of bacteria used in this study (**S8 Table**) have been made available online (https://github.com/ParkinsonLab/eps_biofilms).

To reconstruct putative EPS operons from the sets of loci retrieved from our searches, we first retrieved locus start and stop positions for each locus from their RefSeq entry. We then define putative operons using the following two rules: first only loci that occur within a distance of twice the size of a reference EPS operon to other loci are considered; second intergenic distances of individual loci must be < = 5 kb; third putative operons must consist of at least one locus encoding a putative polysaccharide synthase, together with at least one other locus. To detect previously undiscovered loci that may have been missed in the first rounds of HMM searches, predicted loci of reconstructed operons were used to generate expanded locus-specific HMM models and were subjected to an additional round of HMM searches. This process was performed using custom Perl scripts and results in a list of predicted EPS operons identified in each of the 1733 genomes.

### Classification of evolutionary events

For each EPS system (cellulose, acetylated cellulose, PNAG, Pel, and alginate), the locus assignments of each reconstructed operon was compared to a defined reference EPS operon

compositions and locus ordering (**S5 Table**) and were classified into the following evolutionary events; 1) locus losses—the total number of reference loci missing or not detected by HMM searches; 2) locus duplications–number of distinct loci appearing as multiple significant hits to the same HMM model < 10 kb apart; 3) locus fusions–the number of loci that were significant hits to two or more reference EPS locus HMM models; 4) operon rearrangements–the number of predicted operons with locus ordering (accounting for transcriptional direction) different from the reference operon; 5) operon duplications–number of predicted operons (as defined above) present in the same genome > = 10 kb apart.

## Classification of EPS loci

Systematic classification of each EPS operon family starts with first merging closely related sequences using CD-HIT v.4.6.3 [94] with default settings (using global sequence identity threshold 0.9; word length 5) to generate a non-redundant set of sequences for each family. Multiple sequence alignments (MSAs) were then generated using MUSCLE and trimmed using trimal v.1.2rev59 [95] (using -automated1 setting). The resulting alignment was then used to construct a consensus phylogenetic tree using PhyML v.3 [96], with default parameters (LG substitution model, with 1000 bootstrap replicates). For each consensus tree, pairwise evolutionary distances (defined as the number of expected average number of amino-acid substitutions per site) for all locus protein sequences were extracted using a custom perl script. These evolutionary distances were subsequently used to iteratively generate sets of clusters, with proteins sharing an evolutionary distance less than a defined cutoff (starting at 0 and incrementing by 0.01) placed in the same cluster. This results in the generation of increasingly coarse clusters of sequences with increasing sequence dissimilarity, such that in the final step all sequences are assigned to a single cluster. At this stage, for all possible clusterings three metrics are calculated and summed together to calculate a clustering quality score: (1) proportion of sequences clustered (p) number of sequences clustered / total number of sequences); (2) the average silhouette score (s_avg) [97]:

For each sequence, i, its silhouette score, s(i), is defined as:

$$s(i) = \frac{b(i) - a(i)}{max(a(i), b(i))}$$

Where a(i) = average evolutionary distance (expected number of substitutions per site) i) is the lowest average evolutionary distance to any other cluster of which i is not a member; and (3) Dunn index (DI)[98], for a set of m clusters, its Dunn index, DI, is defined as:

$$DI = \frac{min_{1 \leq i \leq j \leq m} \delta(C_i, C_j)}{max_{1 \leq k \leq m} \Delta_k}$$

Where DI is the evolutionary distance between clusters i and j and Δc is the size of cluster c. Note that a higher s(i) indicates that a sequence is well matched to other members of its cluster and not well matched to neighbouring clusters. Furthermore, a higher DI indicates clusters that are compact (smaller cluster sizes) and well differentiated (larger inter-cluster distances). Thus, the evolutionary distance cutoff which maximizes p + s_avg + DI is chosen as the optimal phylogenetic clustering for a given set of EPS locus sequences.

In these analyses we have chosen not to incorporate bootstrap support parameters. Bootstrap values have previously been used in phylogenetic-based clustering approaches (particularly for epidemiological studies of pathogen transmission and evolution [99,100]) and provide an important metric in assessing the reliability of phylogenetic tree topologies [101]. In contrast, we define clusters in a topology-agnostic manner by clustering sequences based

only on pairwise evolutionary distances. This provides a readily automated approach for defining evolutionary relationships, particularly for sets of sequences that exhibit diverse phylogenetic distributions that are subject to varying evolutionary selection pressures and feature a variety of sequence lengths.

### Construction of EPS operon genomic-proximity networks

To visualize evolutionary and genomic organization relationships of predicted EPS operons, genomic proximity networks were generated in which each node represents an individual EPS locus cluster (as defined above), and an edge connecting a pair of nodes represents the average genomic distance (base pairs) between loci represented by each node found in the same genome. Further, nodes are represented as pie-charts indicating phylogenetic distribution of each EPS locus, as defined by NCBI taxonomic classification scheme. Networks were visualized using Cytoscape (version 3.5) [102].

## Supporting information

**S1 Fig. Lifestyle and niche distribution of predicted EPS operons.** The number of bacterial genomes with different combinations of predicted EPS operons, further represented with their distribution (% bacterial genomes) across different lifestyles and environmental niches. (PDF)

**S2 Fig. Species diversity of predicted synthase dependent EPS systems (shannon diversity).** (PDF)

**S3 Fig. Identification of gram-positive pel operons.** (A) Subnetwork depicting Gram-positive *pel* operon clades with varying numbers of loci identified as significant matches (e-value < 1e-5) in first-pass (unfilled nodes) and iterative HMM searches (grey nodes). Selected examples shown: (i) PelA-PelFG sequences identified by first-pass HMM hits; (i.b) Iterative HMM searches identifying additional *pelA* loci in *B. cereus* ATCC 10987, a known pellicle producing Gram-positive; (ii) Additional *pelD* loci identified by iterative HMM; (iii) Gram-positive *pel* operons with only *pelF* and *pelG* loci identified. (B) Operon organizations of selected examples of Gram-positive *pel* operons (corresponding highlighted in panel A) with additional highly divergent loci identified (red boxes: hits above HMM e-value threshold of 1e-5). (PDF)

**S4 Fig. Genomic-proximity network of phylogenetically clustered *pga* operons.** Phylogenetically clustered operon loci are arranged according to the canonical *pga* operon ordering indicated by the grey sidebar. Inset boxes depict selected examples of *pga* operon clades distinguished by evolutionary events: i) Divergence of *pgaD* corresponding to related enterobacterial species including pathogen-specific losses of *pgaA* and *pgaB* loci critical for PNAG export; ii) Operon duplications occurring in aquatic niche dwelling bacteria, including a partial duplication of the *pga* operon specific to the opportunistic pathogen *Acinetobacter baumannii* spp. and a whole operon duplication identified in *Methylovora versatilis*; iii) A unique *pga* operon organization among environmental bacteria lacking a *pgaD* locus; iv) Gram-positive *ica* operons (annotated by their HMM hits to corresponding Gram-negative *pga* loci) with divergent *icaB* loci, resulting from novel domain acquisitions (iv.b and iv.c); v) A novel *pga* derived operon resulting from multiple tandem duplications of the *pgaC* polysaccharide synthase and lack of detectable *pgaA* outer membrane pore and *pgaD*. Node size indicates the relative number of sequences per phylogenetic cluster; node colouring represents the taxonomic distribution of loci for a given cluster; edges connect clusters which co-occur in the

same genome(s); edge colour indicates the genomic-proximity of loci clusters.
(PDF)

**S5 Fig. Genomic-proximity network of phylogenetically clustered alginate operons.** Phylogenetically clustered operon loci are arranged according to the canonical alginate operon ordering indicated by the grey sidebar. Inset boxes depict selected examples of alginate operon clades distinguished by evolutionary events: Inset boxes depict selected examples of alginate operon clades distinguished by evolutionary events: i) Canonical alginate operon organization with a partial operon duplication event identified in *Pseudomonas resinovorans* 136 resulting in the loss of alginate acetylation machinery (ib–indicated by A*); ii) A distinct alginate operon clade (ii.a-c) identified by rearrangement of acetylation machinery (indicated by B*) as well as HGT events with canonical alginate operon possessing species; iii) Atypical alginate operons involving loss of outer membrane transport loci or portions of acetylation machinery in deep sea dwelling bacteria. Node size indicates the relative number of sequences per phylogenetic cluster; node colouring represents the taxonomic distribution of loci for a given cluster; edges connect clusters which co-occur in the same genome(s); edge colour indicates the genomic-proximity of loci clusters.
(PDF)

**S6 Fig. Genomic-proximity network of phylogenetically clustered acetylated cellulose operons.** Phylogenetically clustered operon loci are arranged according to the canonical acetylated cellulose operon ordering indicated by the grey sidebar. Inset panels identify three acetylated cellulose operons identified in *Pseudomonas* spp. (i) and a single *Bordetella avium* genome possessing a duplicated polysaccharide co-polymerase *wssC* locus (ii—indicated by red asterisk). Node size indicates the relative number of sequences per phylogenetic cluster; node colouring represents the taxonomic distribution of loci for a given cluster; edges connect clusters which co-occur in the same genome(s); edge colour indicates the genomic-proximity of loci clusters.
(PDF)

**S7 Fig. Phylogenetic sequence clustering reflect differences in structural conservation between cellulose synthase complex subunits BcsA and BcsB.** Top panel—Sequence conservation was mapped onto the cellulose synthase complex, BcsA-BcsB (4HG6 –*Rhodobacter sphaeroides* ATCC 17025) comprising sequences from eight species representing distinct cellulose operon clades (Fig 4(i)–4(iv)). Lower panels—structural and multiple sequence alignments indicate a high degree of conservation corresponding to BcsA glycosyl hydrolase catalytic core domain and regions of the cellulose translocation channel (i) and UDP binding sites of the BcsA PilZ domain (ii). In Contrast, low overall sequence conservation is found among the carbohydrate binding and ferredoxin domains (CBD1-2, and FD1-2) of BcsB sequences, except the highly conserved cellulose binding site residing in CBD-2 (iii). The translocated cellulose polymer is indicated in green. BcsA domains identified using PFAM predictions for the *R. sphaeroides* reference sequence, BcsB domains were assigned according to [45]. Multiple sequence alignment was visualized generated using Geneious 10.2.2 (http://www.geneious.com), protein structure was visualized using Chimera 1.11.2 [106].
(PDF)

**S8 Fig. Phylogenetic clustering reveals structural evolution of PNAG PgaB periplasmic modifying enzyme distinguishing gram-negative and gram-postive PNAG operon clades.** A)—Multiple sequence alignment of representative sequences comprising all PgaB phylogenetic clusters. Global sequence conservation compared against *E. coli* MG1655 K12 PgaB, phylogenetic cluster PgaB_G1, indicates presence of polysaccharide deacetylase domain (blue box)

but an absence of glycosyl-hydrolase domain in non-PgaB_G1 sequences. Red arrows indicate phylogenetic group specific N-terminal domain fusions predicted by PFAM searches; C-terminal domain fusions identified (red box) as putative hydrolase domains from BLAST searches. B)—A close up view of sequence conservation of PgaB polysaccharide deacetylase domains with indel events highlighted: green boxes indicate insertions identified in non PgaB_G1 sequences; teal boxes indicate insertions in PgaB_G1 sequence residing in the C-terminal alpha-helix cap (yellow box). C–Crystal structure of *E. coli* PgaB (4F9D) indicating conservation of the deacetylase domain catalytic core. D–Deacetylase domain with indel regions indicated according to the colour scheme described for panel B. E–C-terminal alpha helical cap region of the PgaB deacetylase domain indicating insertions of the PgaB_G1 region that are spatially proximal to an N-terminal region of the hydrolase domain (light purple); comparison of the same regions with PgaB_G1 sequence conservation indicated. Multiple sequence alignment was visualized generated using Geneious 10.2.2 (http://www.geneious.com), protein structure was visualized using Chimera 1.11.2 [106].
(PDF)

**S1 Table. Reconstructed synthase-dependent EPS operons.**
(XLSX)

**S2 Table. Phylogenetic sequence clade assignments of reconstructed synthase-dependent EPS operon loci.**
(XLSX)

**S3 Table. EPS reference operon loci by species and RefSeq accession number.**
(XLSX)

**S4 Table. Previously experimentally characterized EPS producing species; successful identification of a predicted EPS operon by the present study, and additional novel EPS operon possessing strains identified.**
(XLSX)

**S5 Table. Summary of synthase-dependent EPS operon evolutionary events.**
(XLSX)

**S6 Table. Synthase-dependent EPS operon locus clade associations.**
(XLSX)

**S7 Table. NCBI reference complete bacterial genomes and genbank accessions used for synthase-dependent EPS operon reconstruction.**
(XLSX)

**S8 Table. Associated lifestyle and environmental niche metadata for bacterial reference genomes.**
(XLSX)

## Author Contributions

**Conceptualization:** P. Lynne Howell, John Parkinson.

**Data curation:** Cedoljub Bundalovic-Torma, Gregory B. Whitfield, Lindsey S. Marmont.

**Formal analysis:** Cedoljub Bundalovic-Torma, Gregory B. Whitfield, Lindsey S. Marmont.

**Funding acquisition:** P. Lynne Howell, John Parkinson.

**Investigation:** Cedoljub Bundalovic-Torma, Gregory B. Whitfield, Lindsey S. Marmont.

**Methodology:** Cedoljub Bundalovic-Torma, John Parkinson.

**Project administration:** P. Lynne Howell, John Parkinson.

**Resources:** P. Lynne Howell, John Parkinson.

**Supervision:** P. Lynne Howell, John Parkinson.

**Validation:** Gregory B. Whitfield, Lindsey S. Marmont, P. Lynne Howell.

**Visualization:** Cedoljub Bundalovic-Torma.

**Writing – original draft:** Cedoljub Bundalovic-Torma, John Parkinson.

**Writing – review & editing:** Gregory B. Whitfield, Lindsey S. Marmont, P. Lynne Howell, John Parkinson.

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
