## [Decision Letter · Decision Letter 0]

18 Nov 2019

Dear Dr Parkinson,

Thank you very much for submitting your manuscript 'A systematic pipeline for classifying bacterial operons reveals the evolutionary landscape of biofilm machineries' for review by PLOS Computational Biology. Your manuscript has been fully evaluated by the PLOS Computational Biology editorial team and in this case also by independent peer reviewers. The reviewers appreciated the attention to an important problem, but raised some substantial concerns about the manuscript as it currently stands. While your manuscript cannot be accepted in its present form, we are willing to consider a revised version in which the issues raised by the reviewers have been adequately addressed. We cannot, of course, promise publication at that time.

Sincerely,

Mark M. Tanaka

Associate Editor

PLOS Computational Biology

Alice McHardy

Deputy Editor

PLOS Computational Biology

[LINK]

The reviewers all found the manuscript to be a potentially important contribution. They make a number of suggestions for improvement. One reviewer remarks that the novelty of your study resides more in the results than development of a new method. This should be addressed - indeed, if you feel it is appropriate you can re-orient the narrative of the paper. All reviewers raise a number of technical issues which should be addressed with revisions or clarification. The reviewers also point out the need for the senior authors to check the manuscript for language errors. In fact, all authors should carefully read and edit the text.

Reviewer's Responses to Questions

**Comments to the Authors:**

Reviewer #1: The paper by Bundalovic-Torma and colleagues presents a comparative analysis of the operons that encode the enzymes synthesizing five exopolysaccharides: cellulose, acetyl-cellulose, poly-N-acetylglucosamine, Pel (a polymer of mannose, rhamnose, and glucose residues), and alginate (mannuronate/guluronate polymer).

The operons are clustered and compared on the basis of their gene content, phylogenetic distribution, and the likely evolutionary history (gene rearrangement, duplication, loss, fusion,and horizontal gene transfer). This is an interesting and potentially useful work.

However, as could be expected for such an ambitious project, a number of important issues have been either glossed over or relegated to the supplementary files. The paper would benefit from addressing the following points.

Major comments.

1. The paper is entitled "A systematic pipeline for classifying bacterial operons...". Unfortunately, I do not see a clear description of any pipeline. An explanatory figure or a flow chart would be helpful. It would also be useful to highlight which steps of the pipeline result in which results (predictions, discoveries).

2. Given the numerous errors in language (see below for examples), which is unexpected for a Toronto-based team, it is essential that the senior author(s) read and checked the entire text.

2. This ms fails to mention the first publication on the use of operon structure to predict functional coupling: Overbeek et al. 1999, PNAS 96 (6) 2896-2901, https://www.pnas.org/content/96/6/2896. This paper needs to be properly acknowledged.

3. The authors do not explain why they have limited their description of the cellulose synthase operon to just four genes, one of which, bcsZ, actually encodes an inhibitor of cellulose biosynthesis (Ahmad et al. 2016, Microb Cell Fact. 15:177. PMID: 27756305). Traditionally, the reference cellulose synthase operon was the best-studied bcsABCD from Gluconobacter xylinus. As detailed in a recent review (ref. 25), the bcs operon in E. coli and Salmonella consists of at least 6 genes, with three more involved in cellulose modification by phosphoethanolamine (Thongsomboon et al. 2018, Science 359:334-338, PMID: 30232265). Ignoring these extra genes and the additional modification of cellulose simplifies the calculations but results in a biased presentation of the diversity of cellulose synthase operons. At the very least the existence of the extra genes and the associated modification must be acknowledged and the reasons for trimming the gene set carefully discussed.

4. In this paper, any deviations from the reference gene set are described in evolutionary terms (gene rearrangement, duplication, acquisition and loss), implicitly assuming that chosen the reference sets are ancestral operons. Since there is no evidence presented for this assumption being correct for any analyzed operon, all these terms must be qualified and the likely pathways of the evolution of the respective operons discussed without making any unnecessary assumptions.

5. It is not clear whether the trimmed operons actually produce the same compound. The example of Pel biosynthesis by B. cereus is very impressive but it could still be an exception.

6. Supplemental Table 7 is a very useful product of this work. Any chance it could be made available to the community as a separate resource (e.g. a web site), not getting lost among 15 other supplementary files?

Minor comments.

L. 52. "all known synthase dependent bacterial biofilm machineries". Not all, there is also Psl, levan, and phosphoethanolamine cellulose, and probably other biofilm-forming polysaccharides.

L. 67. Strictly speaking, the definition of an operon includes co-regulation. Rephrase.

L. 69. What do you mean by "co-conserved"? Rephrase.

L. 70. Remove "subsequently" (or move to the beginning of the sentence).

L. 71 function isolation -> function in isolation

L. 78-79. "role of evolutionary events on operon structure" Role in or Influence on?

L. 103. "EPS are an important constituent". Rephrase.

L. 117-118. Again, what about the Psl and phosphoethanolamine cellulose?

L. 138. these finding -> these findings

L. 156. Actinobacteridae and Rubrobacteridae are obsolete names for the classes Actinobacteria and Rubrobacteria, see https://www.ncbi.nlm.nih.gov/Taxonomy/Browser/wwwtax.cgi?id=201174&lvl=1

L. 170. "The processes ... is poorly understood."

L. 179-180. " the resulting number ... were assessed"

L. 180. ordering -> order

L. 338. pathogen enterobacteria -> pathogenic enterobacteria

L. 384 and elsewhere. Acetylated cellulose does not need a dash.

In the Supplementary Table 6, the "NCBI Genbank Genome ID" column are actually RefSeq genome IDs. RefSeq is different from GenBank, e.g. the RefSeq entry NC_015671 for Cellulomonas [Cellvibrio] gilvus ATCC 13127 corresponds to the GenBank entry CP002665.

Reviewer #2: The manuscript entitled « A systematic pipeline for classifying bacterial operons reveals the evolutionary landscape of biofilm machineries » by Bundalovic-Torma and colleagues represent an incredible effort to characterize the composition and evolution of the five major EPS operons known across the bacterial diversity.

The authors should be commended by the amount of work performed and the depth of their analysis and all the details provided. Despite the load of work, the results are presented, in most cases, clearly and the figures in the text (and supplements) provide a lot of interesting and insightful information.

Overall, I think this manuscript is of interest to the biofilm and comparative genomics community, it provides novel results and further understanding the evolution of different EPS operons. The reviewer also appreciated the analysis of the genomic results in light of the biological function each protein has. There are however, a few points that I feel should be address prior to acceptance.

Major comments

• Validation of results. My major concern relies on the validation of the EPS predictions. How was this performed? Typically, in a real agnostic way, the database of reference genomes should have been split in two, one should be used for generating the models (HMM profiles and genomic reference organization) and the other one as a validation dataset. Was this done?

Further, there is no evidence that the results obtained were systematically contrasted with any published literature, at least a random subset of those.

I would suggest that the authors validate their data using new genomes from NCBI. (They worked on genomes from 2015, and NCBI has most likely doubled the number of genomes since then).

I am aware that there is an accompanying manuscript validating one novel EPS in Bacillus, but it would provide robustness to this manuscript if a more thorough literature search would be done and potentially detect false negatives, or maybe even false positives.

• L157-158: Statistical analysis. I do not understand how a t-test could have been done with this data. This data concerns distribution of a trait (EPS) across a character (pathogen-non pathogen). This should be analysed using contingency tables and the appropriate statistical analysis would be a Chi-squared or a Fisher’s test but not a T-test. Further, given that there are multiple EPS operons, this should be then corrected post hoc for multiple observations. Statistical analysis should be redone and claims of link towards pathogenicity should be modified accordingly.

• L157-158: In some occasions, pathogenicity is not a well-defined trait. For example, many species are not pathogenic but there can be a reference in which in a given patient they may have caused disease. For example, commensal bacteria can under certain circumstances (immunosuppressed individuals) become facultative pathogens. Were these cases consistently assigned as pathogens or non –pathogen? This should be more explicitly mentioned in the methods sections. Can you explain for example why two very similar S. aureus, “(Sequence type (ST) 59 from an epidemic lineage of community-associated (CA) methicillin-resistant Staphylococcus aureus (MRSA) isolates. Taiwanese CA-MRSA isolates belong to ST59 and can be grouped into 2 )” were classified as pathogenic (SA957) and non-pathogenic (SA40)?

• Figure 1, L 148-157 I believe it would be more informative to provide the percentage of genomes in which each EPS was found rather than absolute numbers (i.e. as shown in Figure S1). I also believe that for visualization purposes, a figure with a tree of the bacterial diversity and the presence/absence of each EPS could be shown in each clade (or better, the degree prevalence (%) of each EPS on each clade).

• The authors present this paper at the beginning as a methodology paper. I think this is an overstatement, and somehow undermines the important results (evolutionary and biological results) presented here. Authors refer to a “novel” method to study the evolution of operons using phylogenetic clustering. I am confused as to what is the precise novelty of this approach. There are a number of papers that have used genomic operon architecture & phylogenetics to study the evolution for instance of secretion systems in bacteria (doi: 10.1371/journal.pgen.1002983, doi: 10.1371/journal.pbio.3000390), as well as stand-alone programs to detect the presence of complex operons in genomes (doi:10.1371/journal.pone.0110726). I would tone down the novelty of the method and focus directly in the evolutionary results.

Minor comments

• L71 word missing? “do not function in isolation”

• L82 more references could be added for example that investigating the divergence between the outer membrane exporter of bacterial capsules and EPS (https://doi.org/10.1016/B978-0-12-394313-2.00007-X, doi: 10.1128/MMBR.00024-08)

• Figure 1D was confusing at first. Wouldn’t it be simpler to do a more traditional co-occurrence matrix? Or a small interaction network? Or maybe modify the legend. I felt it was somewhat counterintuitive to read the table vertically.

• L199-200 It would be of use for the molecular microbiologists to add to the methods the precise genes/proteins used to generate the multiple sequence alignment, and why were those genes chosen. Were those core genes for each operon? Those described in the literature as essential for the biosynthesis? For instance, concerning cellulose operon, in several E. coli natural isolates, the operon is a divergent operon with bcsABZC on one side and bcsEFG on the other side (both of which are essential for cellulose production, doi:10.1111/j.1365-2958.2009.06678.x).

• In Figure 2, given the low quality/resolution of the trees presented in figure 2A, I wonder whether this panel is necessary or it could go to the supplemental material.

• Also in Figure 2, t would be of help to keep the colours constant for each EPS operon across panels B and C.

• Figure C, phylogenetic distance between loci seems to be sometimes really high > 2. What is the unit represented, SNP/site? Can the authors comment on the robustness of their phylogeny? Bootstrap values are never commented, and it would be of use. Further, do authors believe that with distances over 2, these results are trustworthy? I think this grants a small commentary or caveat section in the discussion

• L257-26X It would help the reader if the authors would briefly state the function of each protein in the bscABZC operon clearly, as it was done in the next section for the pel operon for instance.

• L347-350 It is not clear whether these two species are known to produce PNAG or this is a putative novel discovery?

• L 472 Is there a word missing?

• L521 there seems to be a problem with the references. It feels that instead of reference #64 it should be #63. Please check this thoroughly throughout the manuscript.

• L631 Authors mention repeatedly to have analysed 1861 bacteria, but table S6 only contains 1388. This is important, specially to evaluate the relative presence of each EPS operon across clades.

• L631 I think it would be useful to have some summary statistics concerning the bacteria analysed, concerning number of genomes per bacterial clade, gram –ve vs gram +ve bacteria. I am guessing that the NCBI is very biased towards Gammaproteobacteria.

• L641 When building HMM profiles, how was the non-redundancy eliminated (> 97% similarity). Which program or settings were used to determine sequence similarity? This should be detailed in the Methods. Wouldn’t protein clustering be a more appropriate method to reduce redundancy?

• Data availability: I think it would be of use to the community if the HMMs were made available (or if used directly from Pfam: add Pfam link) as well as a simple pipeline for the search to run locally.

Reviewer #3: In this manuscript, the authors state that “a systematic approach for studying the evolution of operon organization is lacking.” They aim to “present a novel method to study the evolution of operons based on phylogenetic clustering of operon-encoded protein families and genomic-proximity network visualizations of operon architectures.”

This is a laudable aim for an important problem.

The authors present methods for analyzing phylogenetic trees to identify clades of functionally similar genes. They then tune their clades using cluster quality measures. This is mostly standard (one might argue about the specific cluster quality measures used, but this is not a major issue). What makes the method more interesting is the joint integration of phylogenetic and genomic-proximity data.

From an algorithmic perspective, there is no question that effective tools and methodologies for these tasks are important, and it is obvious that combining phylogenetic and genomic locus information is essential. As both of these data are graphical, and noisy, the algorithmic challenges are huge. The method that the authors have developed appears to depend on human visual inspection of the operon architectures, and as such, may not be fully automatable for a systematic approach. On the other hand, for such an important task, even small steps towards more effective tools are valuable.

This being said, the paper could benefit from some attention to clarity in the method presentation. This is particularly important as the method is being proposed as a model for others to implement.

For instance, in the subsection titled, “Classification of EPS loci” (lines 671-700) the phylogenetic clustering needs to be clarified. Bootstrap analysis was used, but there is no mention of how (or even if) bootstrap values were incorporated into the method, or even if the final tree analyzed was the consensus tree (or simply the tree constructed from the full multiple sequence alignment). What is meant by “all sequences which share a branch less than the given threshold are assigned to the same evolutionary cluster”? By “branch” do the authors mean tree distance (so that sequences were included in an evolutionary cluster if their tree distance fell within the threshold)?

As a second example, in the iterative HMM methodology described on page 27, lines 637-644, why did the authors select 20 sequences only to expand the HMM training set? Since HMMs generally need large training sets, this is puzzling.

Lastly, although the writing is generally quite good, some editing is needed to fix minor issues in English usage. A few examples are provided:

(a) line 22: “The evolution of these operon-encoded processes is affected by diverse mechanisms such gene duplication, loss, rearrangement, and horizontal transfer.” Insert the word “as” after “such.”

(b) “gene-families” (line 24) should not be hyphenated.

(c) line 88: “the inference of biological function based on sequence similarities of genes or proteins are often…” replace “are” with “is”

**Have all data underlying the figures and results presented in the manuscript been provided?**

Reviewer #1: Yes

Reviewer #2: Yes

Reviewer #3: None

PLOS authors have the option to publish the peer review history of their article (what does this mean?). If published, this will include your full peer review and any attached files.

Reviewer #1: No

Reviewer #2: No

Reviewer #3: No

---

## [Decision Letter · Decision Letter 1]

11 Feb 2020

Dear Parkinson,

We are pleased to inform you that your manuscript 'A systematic pipeline for classifying bacterial operons reveals the evolutionary landscape of biofilm machineries' has been provisionally accepted for publication in PLOS Computational Biology.

Before your manuscript can be formally accepted you will need to complete some formatting changes, which you will receive in a follow up email. A member of our team will be in touch within two working days with a set of requests.

Best regards,

Mark M. Tanaka

Associate Editor

PLOS Computational Biology

Alice McHardy

Deputy Editor

PLOS Computational Biology

The reviewers are now satisfied with the manuscript. Please note the minor inconsistencies raised by Reviewer 2 to be repaired during the production process.

Reviewer's Responses to Questions

**Comments to the Authors:**

Reviewer #1: All my concerns have been addressed.

Reviewer #2: The manuscript entitled « A systematic pipeline for classifying bacterial operons reveals the evolutionary landscape of biofilm machineries » by Bundalovic-Torma and colleagues represent an incredible effort to characterize the composition and evolution of the five major EPS operons known across the bacterial diversity.

I had the chance to review this manuscript in its initial version. The authors have satisfactorily answered/ addressed all the concerns that I raised in my previous review. I believe the manuscript, including the figures, is clearer and has improved significantly. I thus believe this meets the standards of PloS Computational Biology for publication.

Minor comment.

Please revise once again manuscript for minor inconsistencies, for instance, note that the in-file legend of Supplemental Table 8, states it is Supplemental Table 5.

Reviewer #3: My concerns from my original review have been basically satisfied.

**Have all data underlying the figures and results presented in the manuscript been provided?**

Reviewer #1: Yes

Reviewer #2: Yes

Reviewer #3: Yes

PLOS authors have the option to publish the peer review history of their article (what does this mean?). If published, this will include your full peer review and any attached files.

Reviewer #1: No

Reviewer #2: No

Reviewer #3: No

---

## [Editor Report · Acceptance letter]

6 Mar 2020

PCOMPBIOL-D-19-01601R1 

A systematic pipeline for classifying bacterial operons reveals the evolutionary landscape of biofilm machineries

Dear Dr Parkinson,

I am pleased to inform you that your manuscript has been formally accepted for publication in PLOS Computational Biology. Your manuscript is now with our production department and you will be notified of the publication date in due course.

With kind regards,

Sarah Hammond
